
# The wave-age dependent stress parameterization (WASP) for momentum and heat turbulent fluxes at sea in SURFEX v8.1.

Marie-Noëlle Bouin[1,2], Cindy Lebeaupin Brossier[1], Sylvie Malardel[3], Aurore Voldoire[1], and César Sauvage[1,a]

[1]CNRM, University of Toulouse, Météo-France, CNRS, Toulouse, France
[2]University of Brest, CNRS, Ifremer, IRD, Laboratoire d'Océanographie Physique et Spatiale (LOPS), IUEM, Plouzané, France
[3]Laboratoire de l'Atmosphère et des Cyclones, University of La Réunion, CNRS, Météo-France, Saint-Denis, France
[a]now at: Physical Oceanography Department, Woods Hole Oceanographic Institution, Woods Hole, MA, USA

**Correspondence:** Marie-Noëlle Bouin (marie-noelle.bouin@meteo.fr)

**Abstract.** A widely applicable parameterization of turbulent heat and momentum fluxes at sea has been developed for the SURFEX v8.1 surface model. This wave-age dependent stress parameterization (WASP) combines a close fit to available in situ observations at sea up to wind speed of 60 m s$^{-1}$ with the possibility of activating the impact of wave growth on the wind stress. It aims in particular at representing the effect of surface processes that depend on the surface wind according to the

state of the art. It can be used with the different atmospheric models coupled with the surface model SURFEX, including the CNRM-CM climate model, the operational (numerical weather prediction) systems in use at Météo-France and the research model Meso-NH. Designed to be used in coupled or forced mode with a wave model, it can also be used in atmosphere-only configuration. It has been validated in several case studies covering different surface conditions known to be sensitive to the representation of surface turbulent fluxes: i) the impact of a Sea Surface Temperature (SST) front on low-level flow by weak

wind; ii) the simulation of a Mediterranean heavy precipitating event where waves are known to influence the low-level wind and displace precipitation; iii) several tropical cyclones; and iv) a climate run over 35 years. It shows skills comparable to or better than the different parameterizations in use in the SURFEX v8.1 so far.

# 1   Introduction

## 1.1   Background

Turbulent air–sea interactions are known to play a central role in modulating heat and moisture exchanges at interannual to climate scale. They also control the major part of the heat, moisture and momentum exchanges in tropical cyclones (TCs) and, as a consequence, have a strong impact on cyclone intensity (e.g. Emanuel, 2004; Bryan, 2012). Their accurate representation in climate or numerical weather prediction (NWP) models is thus a key step towards better modelling the climate evolution and extreme weather events.

Because the turbulent fluctuations of surface parameters cannot be represented explicitly in atmospheric models, turbulent fluxes are computed using "bulk" parameterizations as functions of mean atmospheric variables at the surface within the





framework of the similarity theory proposed by Monin and Obukhov (1954, MOST). For the wind stress $\tau$, it reads:

$$\tau = \rho u_*^2 = C_d \Delta U^2 \tag{1}$$

with $\rho$ the air density, $u_*$ the friction velocity, $\Delta U$ the difference between the wind speed at a reference level and the surface current and $C_d$ the drag coefficient. Similarly, the heat fluxes are expressed as:

$$H = \rho c_p C_h \Delta U \Delta \theta$$
$$LE = \rho L_v C_e \Delta U \Delta q \tag{2}$$

with $c_p$ the air heat capacity and $L_v$ the latent heat of vaporization. $\Delta \theta$ and $\Delta q$ represent the vertical air–sea gradients of potential temperature and specific humidity, respectively. In neutral conditions and in the surface layer where $u_*$ is supposed to be constant with height, $U(z)$ may be represented as a logarithmic profile:

$$U(z) = \frac{u_*}{\kappa} \log(z/z_0) \tag{3}$$

where $\kappa$ is the von Karman's constant ($\approx 0.4$) and $z_0$ the roughness length. Equivalently, one can write

$$u_* = \sqrt{C_d(z)} U(z) = \frac{\kappa U(z)}{\log(z/z_0)} \tag{4}$$

in neutral conditions and in the absence of surface current.

The roughness length $z_0$ is expressed as the sum of two terms representing the behavior of the surface in (respectively) rough and viscous regimes (Charnock, 1955; Beljaars, 1994):

$$z_0 = \frac{\alpha u_*^2}{g} + \frac{0.11\nu}{u_*}, \tag{5}$$

with $\nu$ the kinematic viscosity of dry air, $g$ the gravitational acceleration, and $\alpha$ the Charnock coefficient. The Charnock coefficient was originally assumed constant but its dependence on wave parameters allows the drag coefficient to vary more explicitly with the sea state. Defining the transfer coefficients $C_d$, $C_h$ and $C_e$ with reasonable accuracy in various conditions of surface wind, stability and sea state has been the subject of a considerable amount of work by many expert teams for at least the last 50 years, and the motivation for many dedicated field campaigns.

## 1.2 Constraints from observations

Direct observations of the turbulent fluxes at sea on buoys, ships and platforms provide constraints on the mean value of the neutral drag coefficient and its growth with wind speed in the range of 10 m wind speed between 5 and 20 m s$^{-1}$ (e.g. Edson et al., 2013). In this wind range, the momentum transferred from the wind to the sea surface is mainly used for the waves to grow up to a well-developed sea, in equilibrium with the wind (e.g. Janssen, 1989, 2004). The part of the wind stress absorbed by the waves has been formulated to be dependent on the stage of development of the wind sea or wave age (defined as the ratio of the





wave peak period to the near-surface wind) by Snyder et al. (1981) and Komen et al. (1984). The wave development, in turn,

impacts the Charnock parameter and roughness length through Eq. (5) and the friction velocity through Eq. (4). Observations

carried out with extreme care and in mainstream conditions (i.e. in the absence of swell or strong surface currents) show indeed

a large variability of the friction velocity and of the drag coefficient at a given wind speed (Fig. 1). Several studies based on

theoretical considerations (Kitaigorodskii, 1965; Janssen, 1989) or field observations (Smith et al., 1992; Donelan et al., 1993)

attribute part of this variability to the effect of the wave growth on $z_0$. Wave steepness (wave height divided by wave length) is

also a good proxy of the sea-state impact on the surface roughness (Taylor and Yelland, 2001). Several parameterizations of the

wind stress with dependence on the wave age have been developed to be used in wind-waves coupled models (e.g. Oost et al.,

2002; Drennan et al., 2003; Janssen, 2004). As a pioneer in the wind-wave coupling domain, the European Centre for Medium-

Range Weather Forecasts (ECMWF) used coupled models for operational forecasts since 1998 and obtained improvement for

surface pressure in medium range NWP and for the 500 hPa geopotential at seasonal scales (Janssen et al., 2001).

For wind speed above 30 m s$^{-1}$, the coupling regime controlling the stress transfer from the atmosphere to the waves is

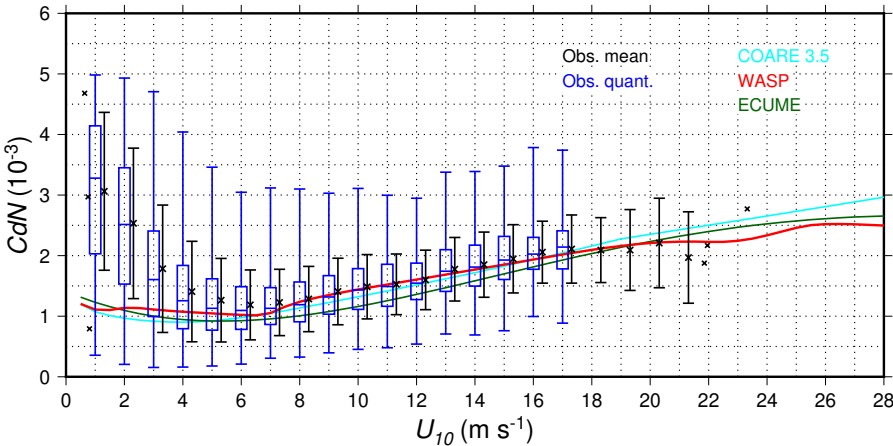

**Figure 1.** Neutral drag coefficient with respect to 10 m wind speed for different parameterizations (COARE3.5, Edson et al. (2013); COARE3.0, Fairall et al. (2003); ECUME Roehrig et al. (2020) and WASP) and in situ eddy covariance observations (see text and Appendix B for details)). The quantiles (blue error bars) correspond to the 10 % – 90 % range.


thought to be less dependent on the wave growth, as most waves are breaking. Direct measurements of wind stress are sparse,

but show no clear dependence on the wave age but a saturation or decrease for wind speeds above 30 to 35 m s$^{-1}$ (Powell et al.,

2003). This saturation itself is confirmed by other (more or less direct) observations (e.g. Black et al., 2007; French et al., 2007;

Jarosz et al., 2007; Vickery et al., 2009; Bell et al., 2012), but the exact corresponding 10 m wind speed where it occurs, the

maximum value of the drag coefficient, and its behaviour at higher wind speeds are still very uncertain. Indeed, all available

estimates beyond 30 m s$^{-1}$ are highly scattered (see Fig. 2 and Fig. B2). This indicates nevertheless that the dependence of the

drag coefficient on the wave growth is not relevant for wind speeds higher than 30 m s$^{-1}$. The saturation or decrease observed

for cyclonic wind speeds must be reproduced in a parameterization (for instance using an analytical function) to match the





observations.

Observations of the heat transfer coefficients show no clear dependence on the wind speed, nor on the sea state. Estimations of

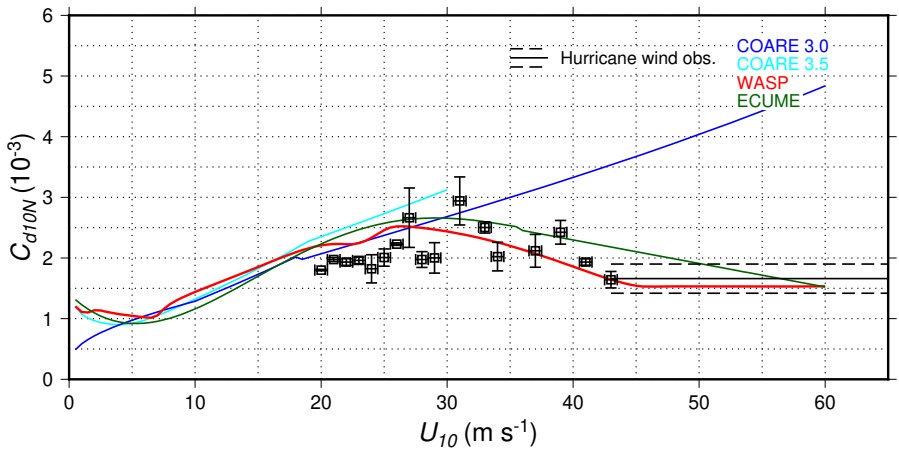

**Figure 2.** Neutral drag coefficient with respect to 10 m wind speed for different parameterizations (COARE3.5, Edson et al. (2013); COARE3.0, Fairall et al. (2003); ECUME Roehrig et al. (2020) and WASP) and summary of observations up to 60 m s$^{-1}$ .


sensible heat flux at sea from sonic anemometers are extremely noisy, resulting in large dispersion between datasets (Fig. 3). Measurements of the latent heat flux is done by gas analyzers, which are very sensitive to rain, high humidity rates at sea, sea spray and pollutants. All of this results in highly scattered values, even in the 5–20 m s$^{-1}$ wind speed range (Fig. 4). However, surface heat transfer play a central role in TC intensification (e.g. Emanuel, 2018) and correctly representing it for strong winds

in NWP models is a key step towards a better forecast of TC intensity. Besides, heat transfer plays a central role on modulating the climate-scale dynamics (in particular in the intertropical band) and can also control local processes even at low winds (e.g. Redelsperger et al., 2019).

### 1.3 Rationale for this work

Several parameterizations of sea surface turbulent fluxes are available in the current SURFEX v8.1 surface model (Masson
et al., 2013), the surface scheme embedded in the atmospheric models used at Météo-France. None of them, however, provides a match to observations for all wind speeds, including the cyclonic conditions, and the possibility to account for the wave growth effect on the roughness length and drag coefficient.

The ECUME parameterization [Roehrig et al. (2020), updated from its initial version, Belamari (2005)] is the default scheme used for operational NWP in the non-hydrostatic, limited-area model AROME (Seity et al., 2011) and in the global model
ARPEGE (Courtier et al., 1991). ECUME is also used in ARPEGE within the CNRM-CM configurations for climate simulations (Déqué et al., 1994). It is also commonly used for case studies with the research-oriented, non-hydrostatic, Meso-NH model (Lac et al., 2018). ECUME has been built by fitting scale parameters for wind, temperature and humidity on observations and enables a close match of the transfer coefficients to observations (Fig. 1 and B2). These transfer coefficients are expressed



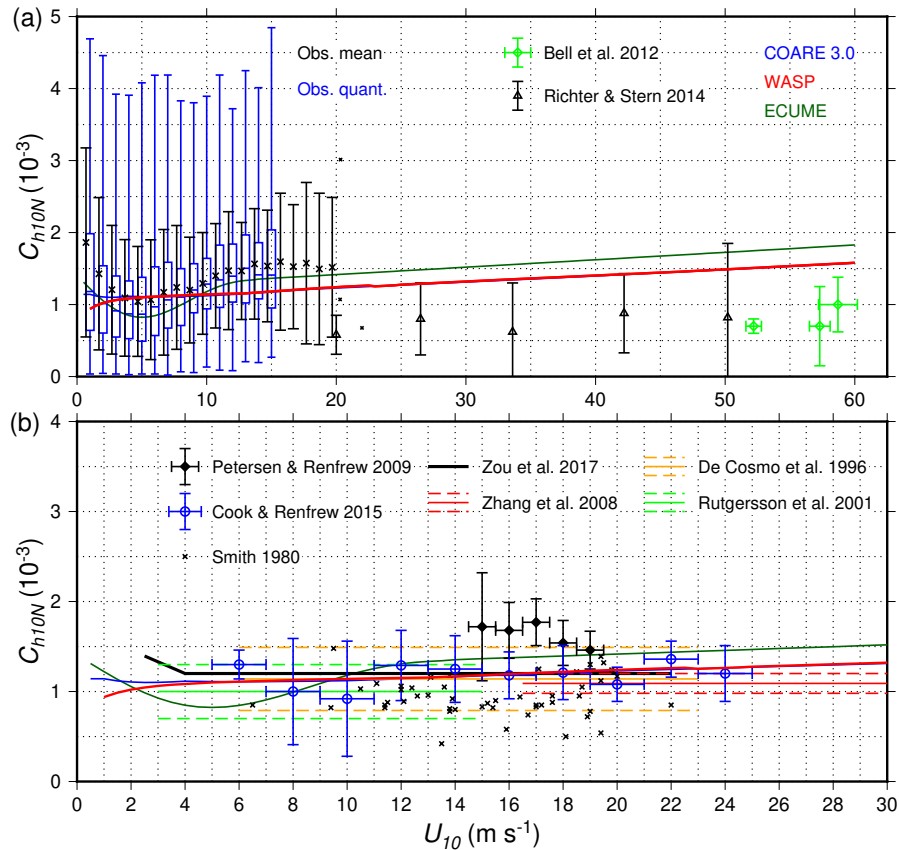

**Figure 3.** Neutral coefficient for sensible heat flux at 10 m with respect to 10 m wind speed for different parameterizations (COARE3.0, Fairall et al. (2003), blue line; ECUME Roehrig et al. (2020), dark green line and WASP, red line) and in situ eddy covariance observations and estimates up to 60 m s$^{-1}$ (a, see text) and additional observations in the wind range 0–30 m s$^{-1}$ (b).

as polynomial functions of the 10 m wind speed only (the roughness length is a diagnostic parameter, Eq. (4) and (5) are not

part of the bulk algorithm).

The COARE 3.0 parameterization (Fairall et al., 2003) can also be used in SURFEX v8.1. It enables representing the impact of sea state on the roughness length through the use of the parameterizations of Oost et al. (2002) or Taylor and Yelland (2001), (Fig. 5b and c). It can also be used in coupled mode with a wave model, the Charnock coefficient (Eq. 5) being computed within the wave model. Using SURFEX with the wave model WAVEWATCH III™[WW3, Tolman et al. (2009)] has been

made possible by the implementation of a surface coupling interface with the OASIS coupler in SURFEX by Voldoire et al. (2017). COARE 3.0 has been fitted to observations of wind stress and heat fluxes in the tropics, for wind speeds up to 18 m s$^{-1}$. It provides a good match with observations of wind stress up to 20 m s$^{-1}$ (Fig. 1) but does not reproduce the decrease of the drag coefficient for winds higher than 30 m s$^{-1}$ (Fig. 5a and 2). As a consequence, it is not suitable for representing the development of TCs or strong storms.

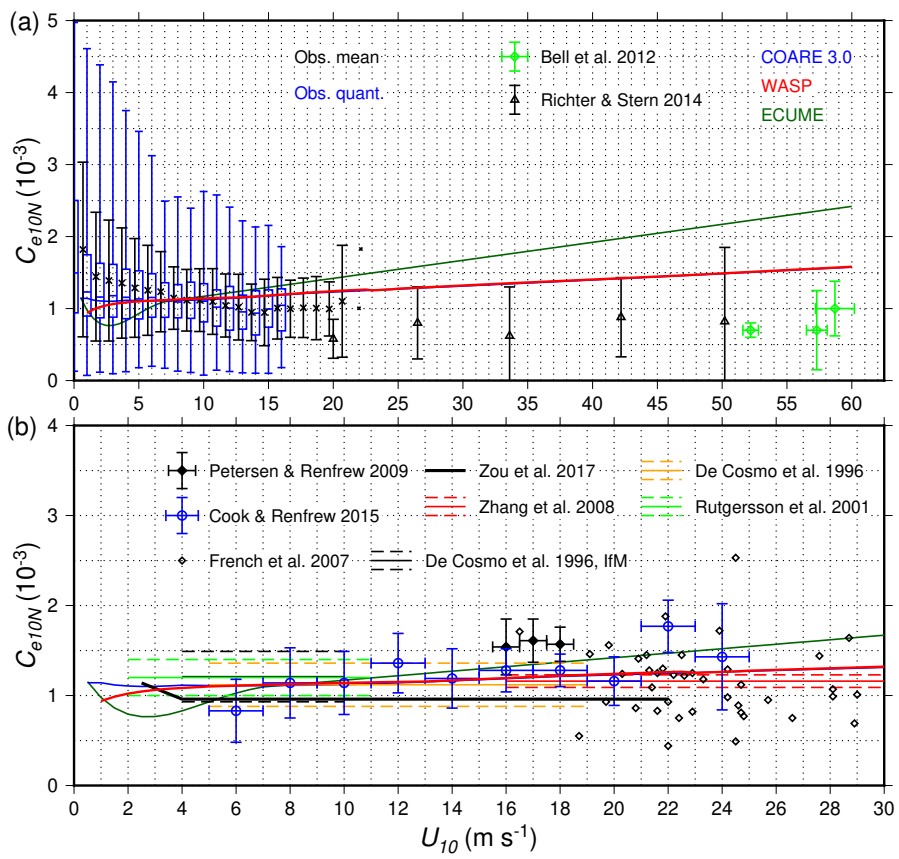

**Figure 4.** Same as Fig. 3 but for latent heat flux.

The new parameterization presented here combines the two aspects of wind–wave coupling and reproducing the decrease of the drag by cyclonic winds. It is based on a very large set of field observations (see Sect. 2.1 for their selection) and ensures that their mean behaviour, in terms of drag and heat transfer coefficients, is well reproduced for wind speeds up to 60 m s$^{-1}$. It is also based on the Charnock relationship with a dependency of the Charnock parameter on the wave age for wind speeds between 7 and 22 m s$^{-1}$, corresponding to the growth of wind sea, (Janssen, 2004),

$$\alpha = A\chi^B \tag{6}$$

where $\chi = c_p/U_{10}$ is the wave age and $A$ and $B$ are polynomial functions of $U_{10}$ (see Appendix A1 for more details). For wind speeds less than 22 m s$^{-1}$, the WASP transfer coefficients closely follow those derived by Edson et al. (2013), using a very large and carefully screened dataset. We do not pretend here to improve much the state of the art of turbulent fluxes at sea that can be used for wind–wave coupling, but rather to design a tool that can be used with every atmospheric model coupled with SURFEX v8.1, producing realistic wind stress and heat fluxes at every wind speed. In addition, the drag coefficient varies as a function of the wave age for a given wind speed in the moderate- to strong-wind range where wave growth is the major process

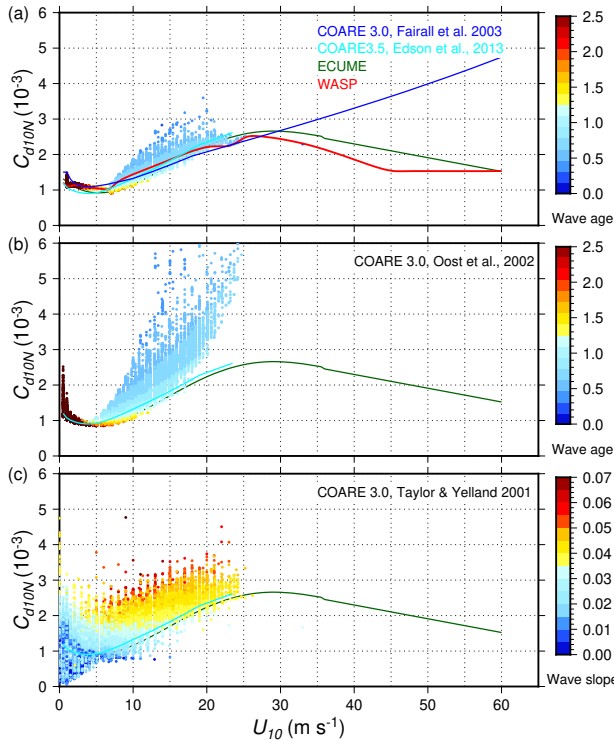

**Figure 5.** Neutral drag coefficient with respect to 10 m wind speed for the parameterizations implemented in SURFEX v8.1 (a) (COARE 3.0, Fairall et al. (2003); ECUME Roehrig et al. (2020), and WASP (present study)), COARE 3.0 with dependence on the wage age (b - color scale, Oost et al. (2002)) and COARE 3.0 with dependence on the wave steepness (c - color scale, Taylor and Yelland (2001)). The COARE 3.5 parameterization (dark green line, Edson et al. (2013)) is shown as the best fit to observation up to 23 m s$^{-1}$. The surface data used to compute the drag coefficient with wave impact in (a, b, c) are the hourly observations of the LION moored Météo-France buoy (centre of the Gulf of Lion) between 2001 and 2014.

absorbing the wind energy.

The next section presents the principle used for building the new parameterization, the observations used to check the mean values of the transfer coefficients for a given wind speed, and the dependency of the drag coefficient on wage age. These options are discussed with respect to the literature and information from various datasets. Section 3 presents the four case studies that were used to validate this parameterization. Some conclusions are given in Sect. 4.

## 2 The WASP parameterization

We present here, first the observations that we retained to fit the mean values of the transfer coefficients, then the transfer coefficients we obtained as functions of the 10 m wind speed. Except specified otherwise, the transfer coefficients developed





in this work corresponds to neutral transfer coefficients at the height of 10 m. They can be expressed as:

$$C_{xN10} = \frac{\kappa^2}{\log\left(\frac{10}{z_0}\right)\log\left(\frac{10}{z_{0x}}\right)} \tag{7}$$

with $x=d$ for wind stress, $h$ for sensible heat and $e$ for latent heat, and where $z_{0x}$ is a roughness length characterizing the surface properties for the given variable. The non-neutral transfer coefficients used in Eq. (1) and (2) are expressed for a given height $z$ as:

$$C_x^{1/2}(\zeta) = \frac{C_{xN}^{1/2}}{\left(1 - \frac{C_{xN}^{1/2}}{\kappa}\psi_x(\zeta)\right)} \tag{8}$$

with $\zeta = z/L$ represents the stability parameter and $L$ the Obukhov length. Neutral conditions correspond to $\psi = 0$, and the Obukhov length is a function of the scaling parameters and values of the wind, temperature and humidity. The stability functions $\psi$ are defined as in Beljaars and Holtslag (1991) with the modifications of Fairall et al. (2003) concerning the free convection conditions (see appendix A2 for their full definition).

WASP is intended to be used either in a coupled mode through the SURFEX v8.1 – OASIS3-MCT coupling interface (Voldoire et al., 2017) or in a forced mode using outputs of a wave model. But using WASP without wave information is also possible. For the latter use, transfer coefficients are functions of the wind speed corresponding to the mean value taken in coupled mode with a well-developed wind sea ("mean values" hereafter), to ensure that the coupled-mode variability corresponds actually to the wave effect. Section 2.1 presents the datasets used to derive these mean values (as explained in Sect. 2.2.1) and the variation

of the surface roughness with sea state is presented in Sect. 2.2.2.

### 2.1 Selection of observations

The parameterization presented here is meant to be used for atmospheric numerical modelling, either operationally with the models of Météo-France of for a large variety of case studies with Meso-NH, with typically the first level at 5 to 20 m above sea level (asl). Whereas it may be tempting to use much finer sampling close to the surface to better represent its influence on

the surface-layer or boundary-layer processes, we believe that doing so within the MOST framework leads to inconsistency (see Pelletier et al., 2021, for a discussion).

The mean values of the transfer coefficients should be representative of a large number of neutral conditions, and the only variability introduced is the impact of the wave age for wind speeds between 7 and 23 m s$^{-1}$ (see Sect. 2.2.2). Turbulent fluxes and transfer coefficients are usually derived from in situ measurements recorded using high-frequency sensors (sonic

anemometers and gas analyzers) with either the eddy-covariance (EC) or the inertial-dissipative (ID) methods. While obtaining reliable estimates using the ID method is easier and more straightforward, it implies strong assumptions on the surface-layer structure, which restrict its use. In this study and for wind conditions up to 25 m s$^{-1}$, we use only carefully checked datasets from measurements at 5 m asl or above computed using the EC method. Thanks to the effort of the observing community, a large number of such datasets exist and many of them were already used by Edson et al. (2013) for deriving the wind stress



parameterization COARE 3.5 (see Table B1 for a list). This results in more than 27 000 individual data (representing 10 to 30 min of measurements each) for $C_d$, 21 000 for $C_h$ and 24 000 for $C_e$. This covers the wind speed up to 22 m s$^{-1}$ for $C_d$, and 20 m s$^{-1}$ for $C_h$ and $C_e$. These observations were binned in intervals of 1 m s$^{-1}$ of wind speed, screened and quality checked. The screening consists in evaluating the symmetry of the binned distributions and whether they correspond rather to normal or log-normal laws. Depending on the results, outliers more than 4 standard deviations from the mean values were removed.


Other historical datasets available in the literature (see Table B2) have been used for the range of wind speed up to 30 m s$^{-1}$. Direct EC measurements in strong winds are scarce and usually made airborne at height between 30 and 500 m asl (Black et al., 2007; Vickery et al., 2009; Cook and Renfrew, 2015). For extreme winds between 30 and 60 m s$^{-1}$, only very few observations are available, especially for the heat transfer coefficients. Some of them are derived from profiles of dropsondes

(Powell and Ginis, 2006), mostly computed indirectly from the effect of the wind stress on the oceanic surface layer, which is more easily sampled than the atmospheric boundary layer in extreme conditions (Jarosz et al., 2007; Hsu et al., 2017; Richter and Stern, 2014). The observations used in this study that correspond to extreme conditions are listed in Table B3.

All these data were used as constraints to derive the transfer coefficients, with different principles for the drag or the heat transfer coefficients, as detailed below.

**2.2   Drag coefficient**

The neutral drag coefficient is first constructed as a mean value, depending on the wind speed only, and fitted to available observations in the wind range from 5 to 60 m s$^{-1}$. Then, a variability depending on the wave age in the wind range of 7 to 25 m s$^{-1}$ is added to the mean value.

**2.2.1   Mean fit to observations**

In the wind range covered by the in situ, EC observations used to derive the COARE 3.5 parameterization (Edson et al., 2013), namely 0 to 21 m s$^{-1}$, the mean value of the neutral drag coefficient is aligned on the COARE 3.5 parameterization, which we consider as the state of the art for drag coefficient. For wind range above 21 m s$^{-1}$, we use data published in the literature (Tables B3 and B2) from less direct measurements, like airborne observations transformed into 10 m wind speed, and measurements on platforms, which may be flawed by the flow distortion. These observations are shown in binned form of

1 m s$^{-1}$ wind speed in Fig. 1 and 2 (see also Fig. B2 for the detail of observations above 30 m s$^{-1}$). Between 25 and 45 m s$^{-1}$, a polynomial function of the wind speed is used to represent the drag coefficient. This function is fitted on the data with weighting based on the uncertainties published with the data (average values and standard deviation are computed with weights equal to the inverse of the variance of the individual datasets). The root mean square of the residuals on $C_d$ is $3.35 \pm 0.32 \times 10^{-4}$. For the wind speed range above 45 m s$^{-1}$, we consider a constant drag coefficient in the continuity of the previous wind

speed range, with a value $1.56 \times 10^{-3}$. The weighted average of the published datasets for the drag coefficient in this wind range is $1.66 \pm 0.24 \times 10^{-3}$, compatible with the value chosen for this constant part of the drag coefficient.





### 2.2.2 Variability with wave growth

We aim here at introducing some variability in the drag coefficient with respect to the wave growth. In his seminal work, Janssen (1989, 1991, 2004) integrated the input term $S_{in}$ in the wave model to derive the part of the wind stress absorbed by the wave growth and to scale the Charnock coefficient. This approach, used for two-way coupling in the operational Integrated Forecasting System (IFS) at ECMWF (Janssen, 2004), is well adapted to operational use with fixed resolutions, and careful tuning and upgrades of the wave and atmospheric model physical parameterizations. Conversely, it is not appropriate for being included in SURFEX v8.1, which is intended to be used with several atmospheric models at various resolution, for NWP, climate or research applications with variable configurations. Indeed, the Charnock parameter computed this way is mainly sensitive to the high-frequency tail of the spectrum (see Eq. 5.22 and 5.24 in Janssen (2004)), which is always parameterized in wave models, because high frequencies cannot be represented explicitly. Some sensitivity tests showed that there is few variability in the Charnock parameter due to the wave field variability for a given wind speed. Thus, the benefit of coupling with a wave model are reduced. The WASP approach used here has two advantages, compared to the Charnock parameter approach: i) it enables to check the validity of the wave parameters used for the coupling (produced by the wave model) against observations; ii) the Charnock parameter is defined differently depending on the range of wind speed considered, enabling to represent in a more physically sound way its behaviour and possible dependency on the waves.

In the wind speed range between 7 and 25 m s$^{-1}$ where the roughness dependency on the wind sea is maximum, the Charnock parameter is expressed as in Eq. (6). Below 7 m s$^{-1}$, the Charnock coefficient is a power function of $U_{10}$, and a polynomial of $U_{10}$ above 25 m s$^{-1}$. The WASP drag coefficient, with a dependence on the wave age, is shown in Fig. 1 and 5a.

## 2.3 Heat fluxes

The principle retained here for building the heat flux transfer coefficient is very similar to the one of the COARE 3.0 (Fairall et al., 2003) parameterization. It is clear from Eq. (7) that the neutral transfer coefficients both for sensible and latent heat fluxes depend only on the roughness lengths $z_{0x}$ and $z_0$. The values of the neutral transfer coefficients for turbulent heat $C_{hN}$ and $C_{eN}$ corresponds to those of the COARE 3.0 parameterization. Then, Eq. 7 is inverted to obtain the value of $z_{0x}$, $z_0$ being obtained in WASP as explained in Sect. 2.2. In the following, we use datasets of available observations to evaluate these parameters for wind speed in the range 0 to 60 m s$^{-1}$. These observations are grouped in direct, EC measurements between 0 and 21 m s$^{-1}$ for $C_{hN}$, 0 and 19 m s$^{-1}$ for $C_{eN}$, and less direct measurements for higher wind speed, available as mean values with estimates of uncertainties for a given wind range or in binned form (see Fig 3 and 4).

### 2.3.1 Sensible heat flux

The direct EC observations have a mean value of $1.388 \pm 0.044 \times 10^{-3}$ for $C_{hN}$ and the high wind or less direct observations (in the range 11–60 m s$^{-1}$) a weighted mean of $1.081 \pm 0.020 \times 10^{-3}$. The mean values are computed as weighted means using the standard deviations of different groups of observations as weights. All together, the whole dataset gives a weighted





mean of $1.143 \pm 0.021 \times 10^{-3}$, very close to the constant value of $C_{hN}$ in WASP. The mean difference and standard deviation between WASP and the binned values of this dataset are $2.1 \times 10^{-4} \pm 3.5 \times 10^{-5}$.

### 2.3.2 Latent heat flux

The direct EC observations have a weighted mean of $1.159 \pm 0.034 \times 10^{-3}$ for $C_{eN}$ and the high wind or less direct observations (in the range 11–60 m s$^{-1}$) a weighted mean of $1.155 \pm 0.012 \times 10^{-3}$. All together, the whole dataset gives a weighted mean of $1.156 \pm 0.011 \times 10^{-3}$, even closer to the constant value of $C_{eN}$ in WASP than for $C_{hN}$. The mean difference and standard deviation between WASP and the binned values of this dataset are $1.3 \times 10^{-4} \pm 3.4 \times 10^{-5}$.

### 2.4 Direct comparison

An offline test was performed to assess the differences between the current version of ECUME used in the Météo-France NWP and climate runs, and WASP. The SURFEX v8.1 model was used to compute the friction velocity and turbulent heat fluxes with either the ECUME or WASP scheme on the same dataset corresponding of observed atmospheric parameters, SST and wave parameters $H_s$ and $T_p$. This dataset consists in more than 53 000 hourly in situ measurements at the Lion buoy, located in the Gulf of Lion, between December 2001 and February 2014. They represent a large range of atmospheric conditions (Fig. 6) with wind up to 25 m s$^{-1}$, air temperature between 5 and 28 °C, relative humidity down to 40 % and wave age ($C_p/U_{10}$) as low as 0.4 due to strong wind and short fetch in mistral conditions. Strong winds in the Gulf of Lion correspond overall to the offshore blowing mistral and tramontane winds, resulting in strongly unstable conditions with dry air, young waves, and significant wave height up to 6 m. Figure 7 shows the difference obtained using WASP rather than ECUME on the fluxes of momentum, sensible and latent heat, as a function of the different surface conditions at the buoy. Warm colours and triangles pointing upward indicate positive differences (the fluxes obtained using WASP are higher than those obtained using ECUME) and blue shades and triangles pointing downward negative indicate negative differences. The comparison of friction velocities obtained using WASP and ECUME (Fig. 7a) shows that the difference does not depend at first order on the wind speed but on the wave age. As expected, young waves give higher friction velocities than older waves. The larger scattering of the difference which is obtained for the lowest and highest wave ages is an artefact due to the smaller size of the sample. For more common conditions, i.e. between 7 and 20 m s$^{-1}$ and wave ages below 1, WASP gives consistently higher friction velocities than EC-UME (8 %). In weaker wind conditions, the difference is not significant.





**Figure 6.** Probability distributions of the data recorded at the Lion buoy: $U_{10}$ (red) and sea level pressure (black,a), SST (black) and air temperature (red,b) and relative humidity (black) and wave age ($c_P/U_{10}$, blue, c). These data are used to force the ECUME and WASP schemes for a direct intercomparison.



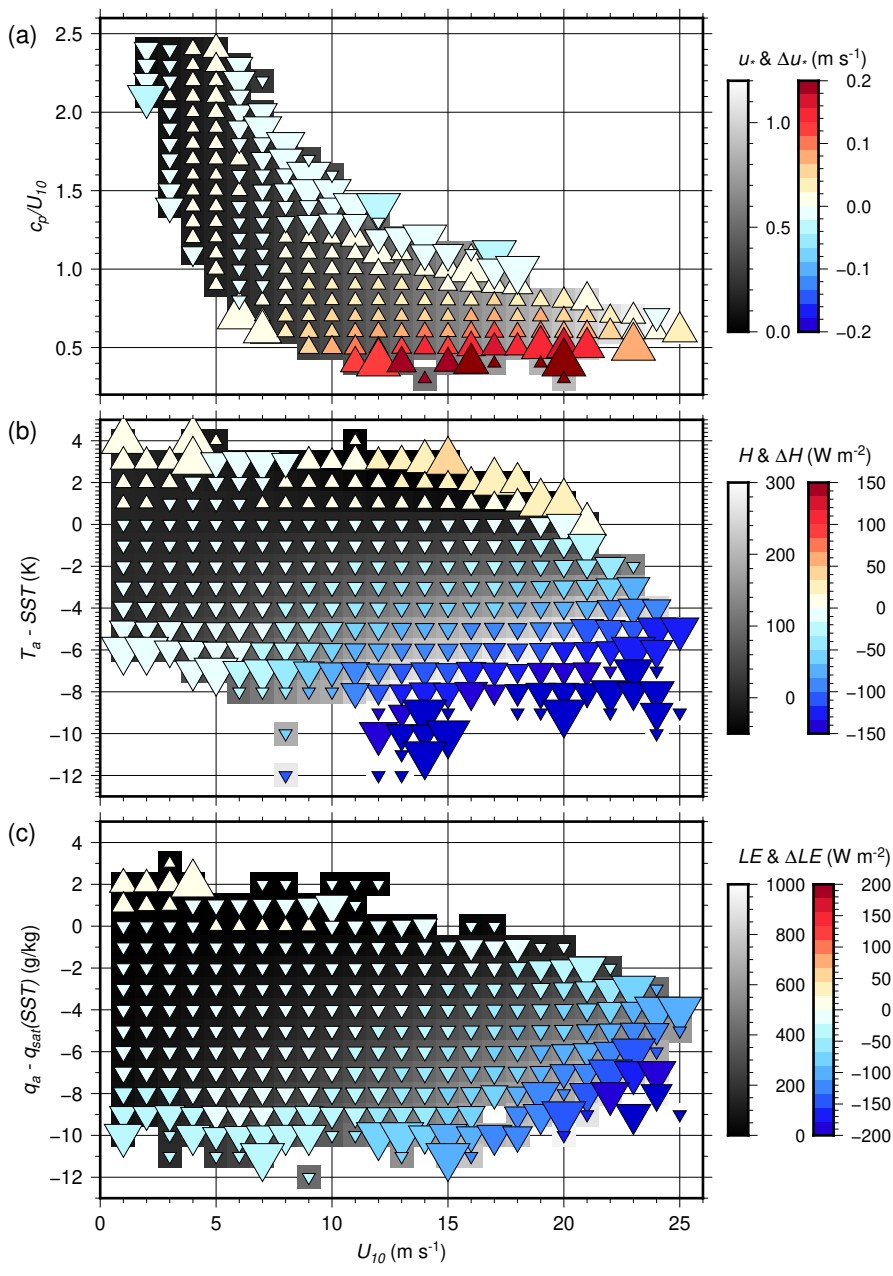

**Figure 7.** Differences between WASP and ECUME (coloured triangles, upward pointing for positive values, downward pointing for negative values) for the friction velocity (a), the sensible (b) and the latent (c) heat fluxes. The differences are mapped as a function of the 10 m wind speed on the x axis and of the wave age (a), the air–sea temperature difference (b), and specific humidity difference (c) on the y axis. For the reference, the values of the parameters obtained with the ECUME scheme are plotted with triangles in a gray scale. The size of the triangles indicates the variability around the mean difference.





Both sensible and latent heat fluxes are generally lower with WASP than with ECUME (Fig. 7b,c). The difference of sen-
sible heat flux is very dependent on the air–sea temperature gradient, especially for winds above 10 m s$^{-1}$. In very unstable
conditions, which are rather common in the Gulf of Lion, the difference reaches $-150$ W m$^{-2}$, for only 30 to 40 W m$^{-2}$ in
stable conditions. The latent heat flux is lower whatever the conditions, except for weak wind, warm and moist conditions that
are rarely met in the Gulf of Lion. It can be expected therefore that heat fluxes will be lower with WASP than with ECUME
when simulating tropical cyclones, at least in the intensification phase with wind speed up to 25 m s$^{-1}$.

## 3 Validation with case studies

A key step of building a parameterization consists in checking its behaviour in representative conditions. To do this, we selected
(i) a case study of weak wind and weak heat fluxes, but where the low-level flow is influenced by the effects of a change of
stratification on the non-neutral drag coefficient; (ii) a strong-wind case where coupling wind and waves is known to influence
the low-level flow and the location of heavy precipitation; (iii) several representative cases of tropical cyclones where both
wind stress and heat fluxes control maximum wind speed and minimum sea level pressure; and (iv) a coarse, atmosphere-only
climatic run where the energetic balance over several decades depends on both the wind stress and heat fluxes in weak to
moderate wind conditions. Cases (ii) to (iv) were performed using the operational models of Météo-France in configurations
close to the operational ones. Case (i) was performed using the research model Meso-NH in the same configuration as in
Redelsperger et al. (2019). Case (iii) was of special importance for building WASP as its results led to the tuning of the
parameterization for wind above 20 m s$^{-1}$, where observations do not provide enough constraints. Among these cases, only
case (ii) explicitly takes into account the wave effect using sea state modelled by WW3, other cases use WASP with wave effect
depending on the wind only.

### 3.1 Weak wind conditions: an Iroise Sea case

The case study of a weak low-level flow across a sharp SST front in the Iroise Sea (Redelsperger et al., 2019, R2019 hereafter)
is used to assess WASP in calm atmospheric conditions, with strong change of atmospheric stratification over a few kilometres.
The configuration used here is the same as in R19, and the reader can refer to this paper for a full description of the case study
and modelling configuration.

### 3.1.1 Atmospheric conditions and modelling configuration

The Ushant SST front is a sharp surface front (3 to 5 °C over $\sim$ 20 km), of barotropic (tidal) origin, which is usually present
from March/April to October in the Iroise Sea and moves of about 5 km throughout the day due to the tidal currents. On the
day of the study (2 September 2011), the low-level wind was 3 m s$^{-1}$ from southwest, crossing the front from the warm to
the cold side with a $\sim$ 45° angle. The 2 m temperature was close to 15 °C, in contrast with the 17 °C or higher SST on the
warm side of the front and 15 °C or lower SST on its cold side, resulting in unstable to neutral atmospheric stratification. The
Meso-NH model was used for a 12 hour simulation with three two-way nested domains with horizontal resolution as fine as





100 m on the central domain covering $45 \times 50$ km across the front. The surface conditions (SST) were provided hourly by a simulation using the Model for Applications at Regional Scales (MARS-3D), zoomed at 500 m (Lazure and Dumas, 2008). The atmospheric initial and boundary conditions of the largest domain were taken from the AROME-France operational analyses at 2.5 km (Seity et al., 2011). In the reference simulation, the surface turbulent fluxes were parameterized using COARE 3.0, which is suitable for the weak wind conditions.

In R2019, it is shown that the impact of the SST front on the Marine Atmospheric Boundary Layer (MABL), although in agreement with published results about its effects and intensity (e.g. Small et al., 2008), differs by the mechanism involved. The sharpness of the front combined with the weak flow results in strong advection, and the process involved here is turbulent mixing rather than pressure gradient. This turbulent mixing is enhanced by a strong contrast of stratification across the front, which increases the non-neutral drag coefficient correspondingly (Fig. 12 in R2019). We check here that the same effects are

obtained by running the simulation using the WASP parameterization instead of COARE 3.0.

### 3.1.2   Results

Figure 8 compares the SST, the difference between the SST and the air temperature, the drag coefficient and the momentum flux along a 35 km profile across the front (see Fig. 9c), from the warm side to the cold side. The decrease of the SST from 17.5 °C to 15 °C (Fig. 8a) produces a strong change in the surface stratification (SST $-T_a$ in Fig. 8b), which results in a

strong decrease of the non-neutral drag coefficient $C_d$ (Fig. 8c) from $1.2 \times 10^{-3}$ to $0.5 \times 10^{-3}$. This induces the corresponding decrease in the momentum flux $\tau$ (Fig. 8d). The striking correspondence of the change of non-neutral $C_d$ with the SST front can be appreciated in Fig. 9a for the COARE3.0 parameterization. The role of this stratification change due to advection across the front in controlling $C_d$ has been established in R2019 and is shown here by the difference between the non-neutral and neutral drag coefficients across the front (Fig. 8c - see also Fig. 9c for a map of the neutral drag coefficient, which is

almost homogeneous on the domain). The simulation using WASP rather than COARE 3.0 gives the same results with a small intensification of the contrast between both neutral and non-neutral drag coefficients across the front (Fig. 8c and d, 9a and b), in link with slightly higher values of the neutral $C_d$ by weak winds, Fig. 1.

In this weak wind situation with strong gradient of surface stratification, WASP behaves similarly to COARE3.0 in reproducing the decrease of turbulent stress from the warm side to the cold side of the SST front.

### 3.2   Moderate to strong wind conditions with waves: a Mediterranean Sea case

The western Mediterranean region is regularly affected by heavy precipitation events (HPEs) that are characterized by a large amount of rainfall over a small area in a very short time (typically more than 100 mm in less than one day, Ducrocq et al., 2014; Khodayar et al., 2021). These events regularly lead to flash flooding that is a major threat in the area, as it often causes severe damages and in some cases casualties (e.g. Llasat et al., 2013). At low level, strongs wind with high SST as generally

encountered in autumn govern heat transfer, which moistens and warms the air parcel, thus increases the instability and finally intensifies the convection (e.g. Stocchi and Davolio, 2017; Rainaud et al., 2017; Senatore et al., 2020). The SST fine scale structures and fronts in the Mediterranean are also known to play a role on low-level wind convergence (Meroni et al., 2018,



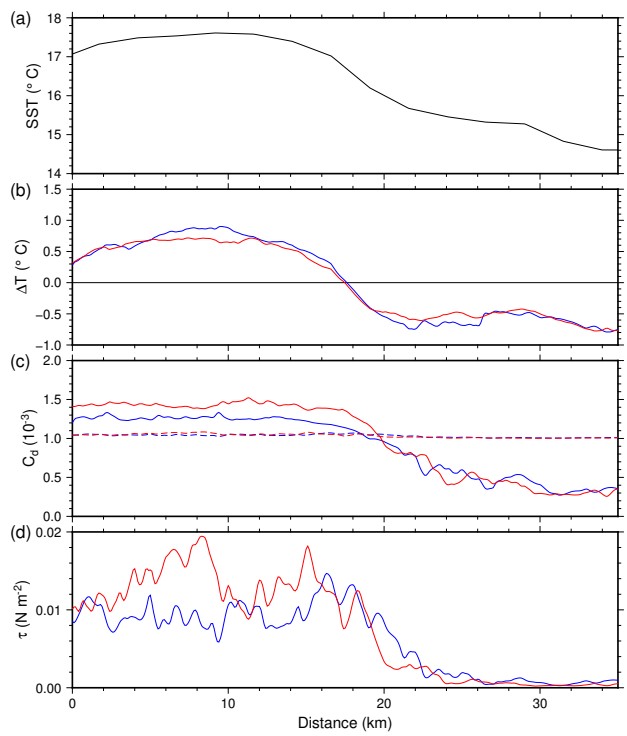

**Figure 8.** Mean values of the SST (a), $\Delta T = \text{SST} - T_a$ (b), non-neutral drag coefficient (solid) and neutral drag coefficient (dashed) (c), and turbulent stress (d) at 12 UT on 2 September 2011 on the Iroise Sea across the SST front, with COARE3.0 (blue) and WASP (red) parameterizations.

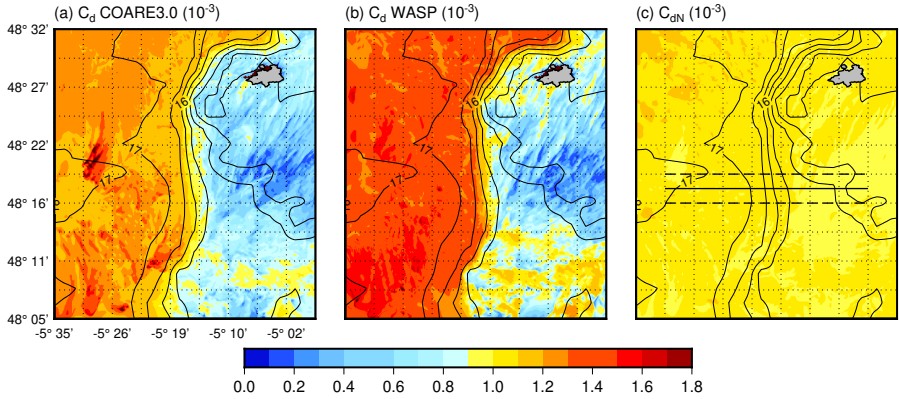

**Figure 9.** Maps of the COARE3.0 (a), WASP (b) and WASP neutral (c) drag coefficient at 12 UT on 2 September 2011 on the Iroise innermost domain (colour scale). The isolines indicate the SST. The black horizontal lines indicates the transect used for extracting the values of Fig. 8.



2020) that is a key triggering mechanism for deep convection and HPEs at sea. Several case studies using kilometer-scale atmospheric models also showed that taking into account the modulation of the surface roughness by the waves can slow down

the low-level flow, shifting the convergence lines and/or modifying the spreading of the cold pool formed below the convective system by precipitation evaporation (e.g. Thévenot et al., 2016; Bouin et al., 2017).

The WASP parameterization has already been tested with and without wave effect by Sauvage et al. (2020, hereafter S2020) on a HPE occurring in mid-October 2016 in South-Eastern France. The wave parameters used as input of the parameterization came from the wave model WW3 in forced or coupled mode. In this case, the wave impact on the surface roughness reduces

the low-level wind speed of more than 1 m s$^{-1}$ over a large area, and a displacement of the HPE of 40 km towards sea. Since the sensitivity of the wave impact within WASP was already investigated in S2020, we use here the same case study and configuration to test the effect of using WASP in the AROME model with respect to the parameterization ECUME currently used for operational forecasts. We first give a short summary of the configuration used and present then the results of the comparison. The WASP parameterization has already been tested with and without wave effect by Sauvage et al. (2020,

hereafter S2020) on a HPE occurring in mid-October 2016 in South-Eastern France. The wave parameters used as input of the parameterization came from the wave model WW3 in forced or coupled mode. In this case, the wave impact on the surface roughness reduces the low-level wind speed of more than 1 m s$^{-1}$ over a large area, and displaces the HPE of 40 km towards sea. Since the sensitivity of the wave impact within WASP was already investigated in S2020, we use here the same case study and configuration to test the effect of using WASP in the AROME model with respect to the parameterization ECUME

currently used for operational forecasts. We first give a short summary of the configuration used and present then the results of the comparison. The WASP parameterization has already been tested with and without wave effect by Sauvage et al. (2020, hereafter S2020) on a HPE occurring in mid-October 2016 in South-Eastern France. The wave parameters used as input of the parameterization came from the wave model WW3 in forced or coupled mode. In this case, the wave impact on the surface roughness reduces the low-level wind speed of more than 1 m s$^{-1}$ over a large area, and a displacement of the HPE of 40 km

towards sea. Since the sensitivity of the wave impact within WASP was already investigated in S2020, we use here the same case study and configuration to test the sensitivity of replacing the operational scheme ECUME by WASP. We first give a short summary of the configuration used and present then the results of the comparison.

### 3.2.1 Case study and modelling configuration.

The complete description of the case study and the AROME model in the configuration used here is given in S2020.

The AROME domain configuration is the one used operationally at Météo-France, and known as AROME-France (Brousseau et al., 2016) with a grid resolution of 1.3 km and 90 $\eta$-levels with the first level at 5 m asl. To assess the sensitivity of the simulated event to a change of turbulent flux parameterization, we performed two identical sets of simulations using either ECUME or WASP with wave forcing from an offline WW3 simulation. Each set was composed of forecast simulations starting at 00:00 UTC on the 12, 13 and 14 October from AROME operational analyses and lasting 42 h. Hourly boundary conditions were

sourced from the ARPEGE operational forecasts (Courtier et al., 1991) except for the SST, which came from the global daily analysis of the Mercator Ocean International (1/12° resolution, PSY4/GLO12 system, Lellouche et al., 2013).





The situation at low level is characterized by a cyclonic circulation that induced a south-easterly flow across the Western Mediterranean Sea and by a strong easterly flow originated from Southern Alps that triggered large sea-surface heat exchanges over the Ligurian Sea and along the French Riviera due to strong wind (up to 20 m s$^{-1}$ observed at the Azur buoy [7.8° E, 43.4°
N]) and to large air–sea gradients. The convergence zone between the warm and moist southerly flow and the dry and cold easterly flow was found to trigger convection over the sea. A second convective system, over Hérault in the South of France, was initiated by an orographic uplift and was fed by the easterly flow. Both systems produced large amounts of precipitation.

The Gulf of Lion was initially affected by the rapid easterly flow, producing a young sea with significant wave height ($H_s$) up to 6 m and strong air–sea fluxes. As the system moved eastwards with the highest wind intensity, the sea state evolved in
time from a well-developed sea to swell in this region. Throughout the event, the French Riviera was affected by strong easterly wind generating wind sea.

### 3.2.2   Results.

The expected impact of parameterization change from ECUME to WASP on this case study is twofold: first, as in S2020, increasing the mean value of the drag coefficient in the range of wind speed [7–20 m s$^{-1}$ and adding variability for a given
wind speed should decrease the low-level wind; and second, the turbulent heat fluxes should be lowered with respect to the ECUME parameterization possibly lowering the convection at sea.

Figure 10 shows that at 14:00 UTC on 13 October (i.e. at the peak of precipitation intensity), the 10 m wind speed actually decreases by 1 to 2 m s$^{-1}$ over a large area in the Ligurian Sea with WASP. The decrease (and local increase) observed in the Gulf of Lion are due to the westward displacement and enhancement of the convergence zone at sea, as observed in S2020. On
the Ligurian Sea which is also the place of strong evaporation, the surface enthalpy flux is significantly decreased by 200–250 W m$^{-2}$ in the WASP simulation (Fig. 11). These two effects have competing impacts on the convective system all along its lifecycle. In ECUME, the stronger easterly wind tends to displace the convergence zone westwards. But, progressively, the larger heat fluxes lead to a more intense convective system at sea. It induces the development of a well marked cold pool below the system that reinforces the convergence line and pushes it eastwards. As a result of these competing effects, there
is no shift of the precipitation area at sea between WASP and ECUME simulations, conversely to what was obtained when comparing simulations done using WASP with and without wave effect in S2020. The convergence and convective system are more stationary, the intense rainfall patch is thinner but the maximum amount of rainfall is quite similar as shown by the accumulated rain amounts between 06 and 12 UTC and between 12 and 18 UTC on 13 October (Fig. 12). For precipitation that hit the Hérault region, we found a small decrease in the rainfall intensity with WASP, in particular during the mature phase of
the system (Fig. 12b,d), induced by the lower warming and moistening of the easterly low-level jet that feeds the convective system.

The WASP parameterization used here forced by realistic sea states produced by a WW3 simulation gives results very comparable to the operational simulation. The predictability of the event was good in general, especially concerning the precipitation over the Hérault region, and WASP enables to obtain similar results with a more realistic sea-surface roughness representation.



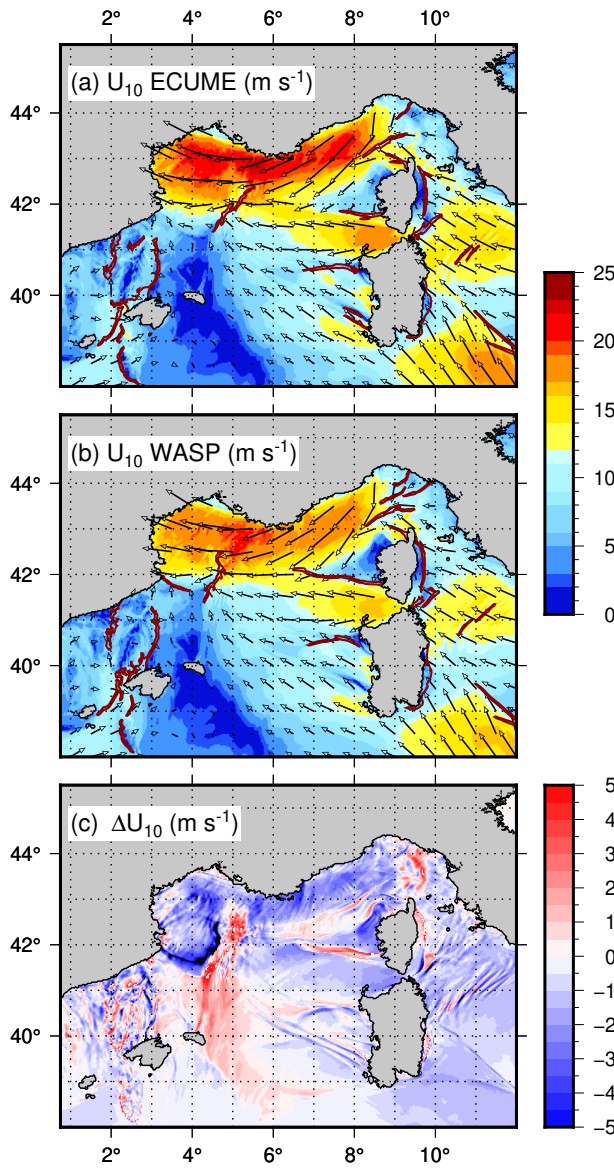

**Figure 10.** Maps of the 10 m wind in the ECUME (a) and WASP forced simulation (b), and difference WASP - ECUME (c) at 14:00 UTC on 13 October 2016. The main convergence area are shown with dark red lines (threshold $10^{-3}$ s$^{-1}$).

### 3.3 Extreme wind conditions: tropical cyclone

WASP is designed to ensure the representation of the variability due to the wave growth and the saturation of the drag coefficient in case of cyclonic winds. The values of the transfer coefficient for heat are reasonably constrained by the observations for winds up to 20 m s$^{-1}$ but between 20 and 60 m s$^{-1}$ observations are too sparse for a robust fit. Case studies of tropical cyclones



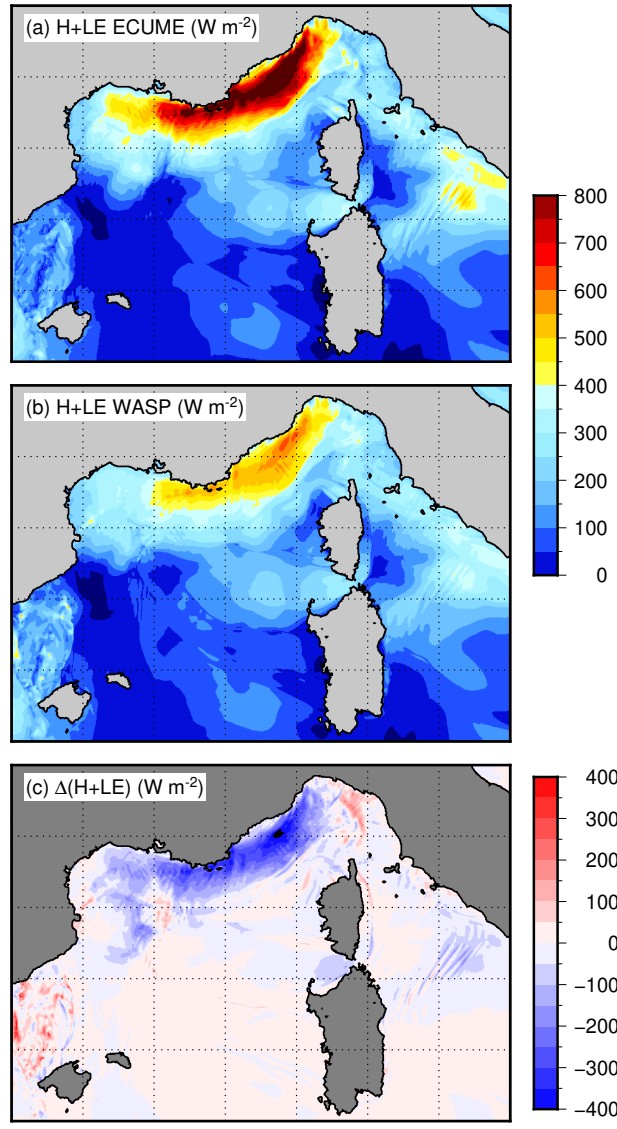

**Figure 11.** Maps of the enthalpy flux in the ECUME (a) and WASP forced simulation (b), and difference WASP − ECUME (c) at 14:00 UTC on 13 October 2016.

can help to validate indirectly the values chosen for the drag and heat transfer coefficients in the wind speed range with no
observations or observations with large uncertainties.



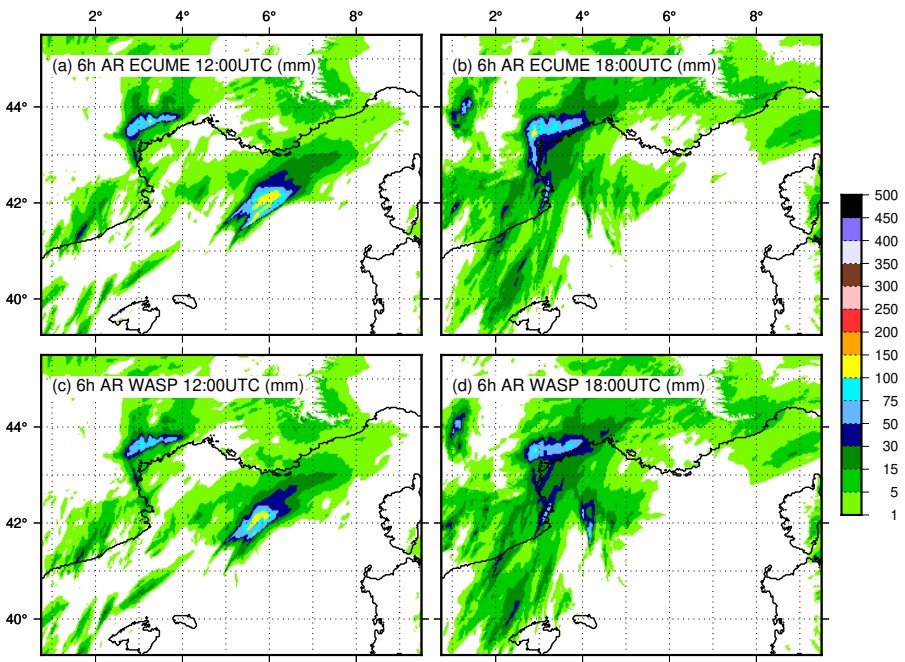

**Figure 12.** Maps of the 6 h accumulated rain in the ECUME (a,b) and WASP forced simulations (c,d) at 12:00 (a,c) and 18:00 UTC (b,d) on 13 October 2016.

### 3.3.1 Case study and modelling configuration.

To test the sensitivity to the turbulent fluxes, we used the current operational configuration of AROME for the forecast of the tropical cyclones in the Indian Ocean (AROME IO hereafter, Bousquet et al., 2020). AROME is used over a large domain centered at 50° E covering Madagascar and the Mozambique Channel. The horizontal resolution is 1.3 km with 90 vertical

levels. It is coupled every 300 s with an oceanic 1D model based on the development of Gaspar et al. (1990), with a prognostic equation of the turbulent kinetic energy with a 1.5 order closure. This 1D ocean model is initialized by the Mercator Océan International global operational forecasts (one hour average) available 6 hourly with a resolution of 1/12°(Lellouche et al., 2018). The surface turbulent fluxes are parameterized by ECUME in the control run (operational configuration), with WASP without waves in the sensitivity experiment. The case studies chosen for this validation are those of the cyclonic season 2021-

2022, with a focus on Batsirai. Batsirai developed at the end of January 2022, reached category 4 on the 02 February 2022 right before hitting La Réunion Island and slightly weakened to category 3 before landing on the eastern coast of Madagascar where it caused a lot of damages. Simulations of Batsirai started at 00:00 and 12:00 UTC on 03 February 2022, and 00:00 UTC on 04 February 2022 and lasted 72 hours. The profiles shown in Figure 13 are composites built from these three runs and ranges of 39 h for the first initial time, 27 h for the second and 15 h for the last one. The output time is 15:00 UTC on the 04 February

matching the time of the Sentinel-1A SAR data at 15:03 UTC on the same day. These SAR high-resolution wind products





are obtained from the IFREMER/Cyclobs database and produced with SAR wind processor co-developed by IFREMER and CLS (Mouche et al., 2017).

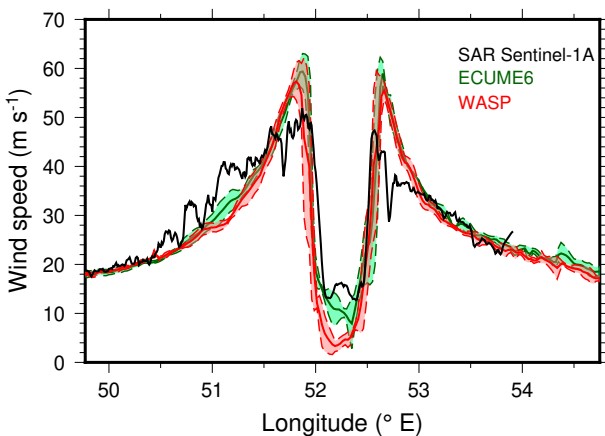

**Figure 13.** West-east composite profiles of $U_{10}$ (m s$^{-1}$) across the AROME IO simulations of Batsirai at the time of SAR Sentinel-1A measurements (15 UTC on 04 February) from operational runs at 3 different initial times. The plain line represents the mean value, the dotted lines the standard deviation.

### 3.3.2 Results

The scores based on the comparison of the minimum of sea-level pressure (SLPmin) and surface maximum wind (Vmax) produced by the simulations with those of the Best Tracks have been produced for three major cyclones of the 2021-2022 cyclonic season in the Southern Indian Ocean (Fig. 14). The Best Track (BT) is the result of the objective analysis of the Regional Specialised Meteorological Centre for Tropical Cyclones of La Réunion, and is considered as the reference in this study. The scores used here aggregate the outputs of about 25 runs with different initial times for every cyclone, either from IFS, AROME IO using ECUME or AROME IO using WASP. AROME IO with ECUME compares well with the BT at forecast ranges up to 12 h but overestimates the cyclone intensity (lower SLPmin and higher Vmax) at longer ranges, even more so at increasing forecast ranges, while IFS overall underestimates the cyclone intensity. AROME IO with WASP underestimates Vmax in the first 12 h (probably due to the effect of the initial conditions) but gives the closest values of SLPmin and Vmax to the BT for the forecast ranges longer than 12 h. For the case of Batsirai where SAR observations are available close to its peak of intensity, direct comparisons of composite 10 m wind speed with SAR surface wind show that the wind speed along a profile across the cyclone are slightly better represented using WASP than ECUME (Fig. 13 and 15).

Simplified, axisymmetric representations of tropical cyclones make the maximum potential intensity directly depend on the ratio of the enthalpy transfer coefficient ($C_k$, analog to $C_e$ here) by the drag coefficient. The minimum value of this ratio, $C_k/C_d$, able to produce maximum surface winds of 45 m s$^{-1}$ or more as currently observed in cyclones of Category 5 was thought to be 0.75 (Emanuel, 1995). These considerations, however, have been contradicted by in situ and wave tank



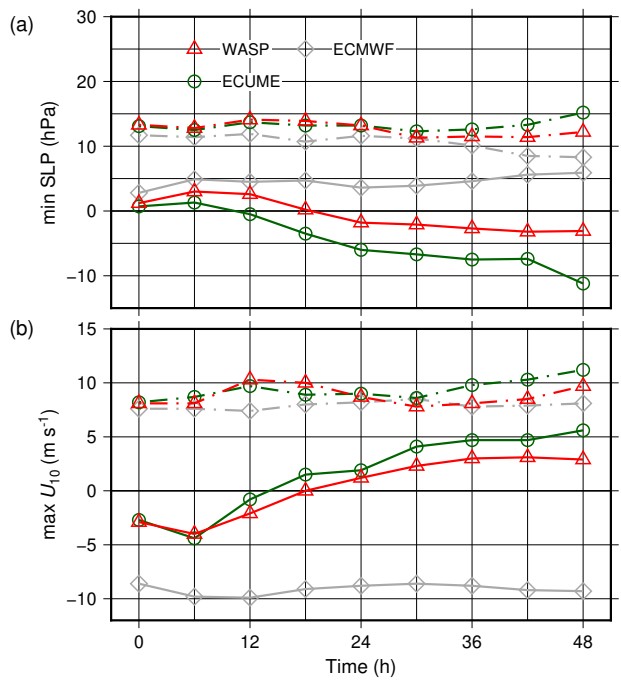

**Figure 14.** Mean bias (solid line) and RMSE (dot-dashed line) for the Batsirai, Emnati and Dumako simulations using AROME IO with ECUME (dark green, circles) or WASP (triangles, red) and IFS (diamonds, gray) for SLPmin (a) and Vmax (b).

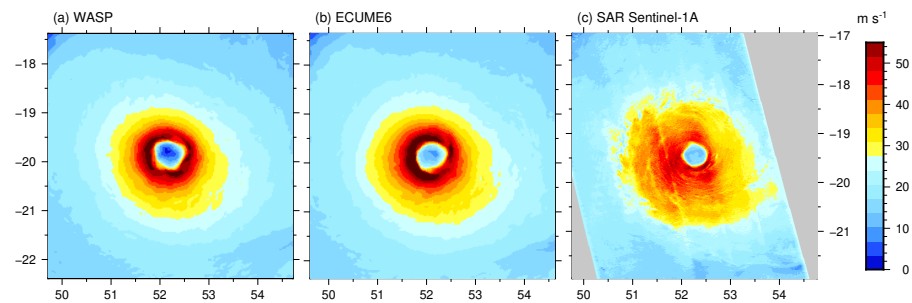

**Figure 15.** Maps of the 10 m wind speed close to the time of maximum intensity on the TC Batsirai, as simulated with AROME IO using ECUME (a), WASP (b) and in the Sentinel-1A SAR product (c). Simulation products shown here are composites from outputs at 15:00 UTC of 3 simulations starting at 00 UTC on 03 February, at 12 UTC on 03 February, and at 00 UTC on 04 February 2022. The time of the SAR observations is 15:03 UTC on 04 February 2022.

observations: increasing surface wind up to 40 m s$^{-1}$ are consistent with a slow but continuous decrease of the $C_k/C_d$ ratio down to 0.5 (Powell et al., 2003; Haus et al., 2010). Recently, simulations based on realistic, high resolution numerical models showed that the $C_d$, $C_k$ values leading to cyclone intensities close to observations and compatible with observations of turbulent fluxes in strong wind actually result in $C_k/C_d$ ratio close to 0.5 (Green and Zhang, 2013, 2014; Nystrom et al., 2020). In such a

respect, the ratio of enthalpy and drag coefficient obtained in WASP stays between 0.4 and 1.0 for wind speeds between 10 and
60 m s$^{-1}$ (Fig. 16). It constitutes a good trade off between the continuous decreasing values given by COARE 3.0 and COARE
3.5, and the values of ECUME increasing probably unrealistically up to 1.5 for surface winds of 60 m s$^{-1}$ and encourages to
test in a more comprehensive way the use of WASP for tropical cyclone prediction.

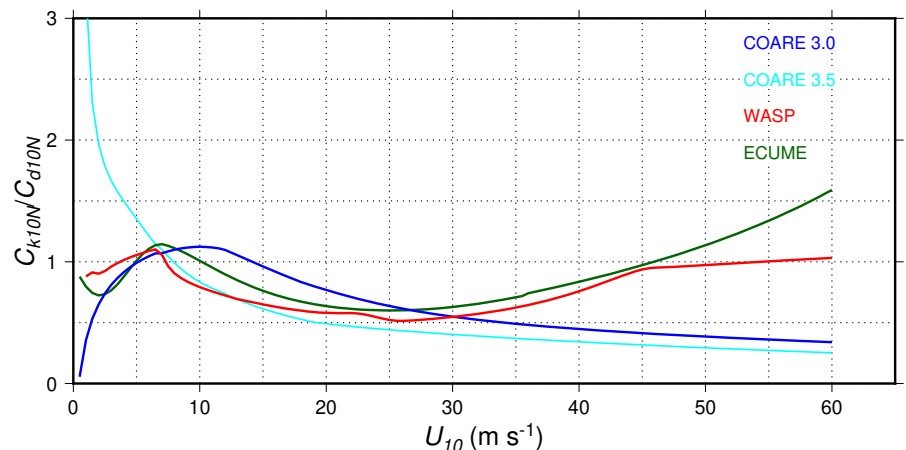

**Figure 16.** Ratios of the neutral enthalpy transfer coefficient and drag coefficient with respect to 10 m wind speed for different parameterizations (COARE3.5, Edson et al. (2013); COARE3.0, Fairall et al. (2003); ECUME Roehrig et al. (2020) and WASP) .

### 3.4 Climate-scale simulation

The sensitivity of climate-scale runs to the turbulent fluxes parameterization was tested in climate mode using the CNRM-CM
model (Roehrig et al., 2020).

### 3.4.1 Configuration

The test has been made in an atmospheric simulation where SST are prescribed on a monthly basis over the 1979–2014
period following the AMIP protocol. The reference simulation for which the air–sea fluxes are calculated using the ECUME
parameterisation has been published in the CMIP6 data base and is extensively described and assessed in Roehrig et al. (2020).
Here we only provide minimal information on this configuration: the horizontal resolution is close to 1.4° and there are 91
vertical levels in the atmosphere with the first level at 10 m. To test the WASP parameterisation, a sensitivity experiment has
been performed were WASP is activated instead of ECUME over the same 35 years (1979–2014).

### 3.4.2 Results

Mapping the differences of surface parameters and fluxes obtained with the WASP and ECUME parameterizations shows an
overall impact of the change of transfer coefficients. In the regions of high annual mean values of heat fluxes, namely the





intertropical basins, the impact of such parameterization changes was explored by Torres et al. (2019). In the present study, $C_{eN}$ in WASP is higher than in ECUME for 10 m wind speed below 8 m s$^{-1}$ and lower for 10 m wind speed above 8 m s$^{-1}$. This results in higher evaporation in the intertropical basins (Fig. 17a,b) with annual mean values in the region between 20° S and 20° N of 121.7 W m$^{-2}$ with ECUME and 123.2 W m$^{-2}$ with WASP. With respect to the interannual variability over 36

years, this change is not significant (at 95 % uncertainty with a Student test). It nevertheless results in overall higher humidity on the ocean (+0.21 g kg$^{-1}$ – not significant, Fig. 17c). Also, stronger precipitation (below the significance level) are obtained along the intertropical convergence zone (ITCZ, Fig. 17d, +0.65 mm d$^{-1}$). Outside the intertropical region, using WASP rather than ECUME results in lower specific humidity near the surface (Fig. 17c) and less precipitation (Fig. 17d). These results are qualitatively similar to those of Torres et al. (2019), see for instance their Fig. 4-2 for the difference between AREF and ACTN.

The lowest level atmospheric temperature annual mean increases slightly in the intertropical regions and decreases at midlatitudes (mostly not significant, Fig. 18c). This is due to the strong changes of the sensible and latent heat transfer coefficient in WASP (overall significant, Fig. 18a, 17a) which impact the sensible and latent heat fluxes, through a decrease at midlatitudes and an increase in the intertropical band (not significantly, Fig. 18b, 17b). Note that stronger decrease on the western boundary energetics areas is partly due to a larger decrease of the heat transfer coefficients by stronger wind.

Finally, the neutral drag coefficient is higher in WASP than in ECUME, whatever the wind speed below 19 m s$^{-1}$ (Fig. 19a). The wind stress is higher everywhere at the sea surface except locally in the Arabic Sea and in the Southern Ocean (Fig. 19a,b, +2.3 10$^{-3}$ N m$^{-2}$). This results in an overall decrease of the wind speed, with stronger effect in the Southern Ocean where the increase of the drag coefficient is the strongest (Fig. 19c, $-0.09$ m s$^{-1}$).

Overall, testing WASP in a climate-scale configuration does not alter significantly the mean climate simulated. No significant

change is obtained except a slight increase of the precipitation in the ITCZ, and a slight cooling and drying effect outside the ITCZ. Further tests should be done in ocean coupled mode to assess the coupling feedback that could arise when switching to WASP air–sea flux parameterization.

## 4 Conclusions and perspective

The WASP bulk parameterization for surface turbulent fluxes has been built based on existing, reliable parameterizations like

COARE 3.0 (Fairall et al., 2003), COARE 3.5 for the momentum flux (Edson et al., 2013), and ECUME (Belamari, 2005; Roehrig et al., 2020). It combines the possibility of representing the effect of the wave growth on the wind stress with transfer coefficients close to field observations at every range of wind speed. It has been developed in the SURFEX v8.1 surface model (Masson et al., 2013) and will be distributed as part of the next official SURFEX release (v9).

In the present study, we assess its behaviour in several case studies performed with the different atmospheric models in use

at Météo-France that can be coupled with SURFEX. It proves to perform reliably with respect to existing parameterizations in various conditions of wind and heat transfer, and to enable an accurate representation of several surface processes. In the case of the Ushant SST front (Redelsperger et al., 2019), the sharp change of stratification along the low-level flow from the warm side to the cold side of the front is well reproduced and leads to a strong decrease of the momentum flux. As a result, the




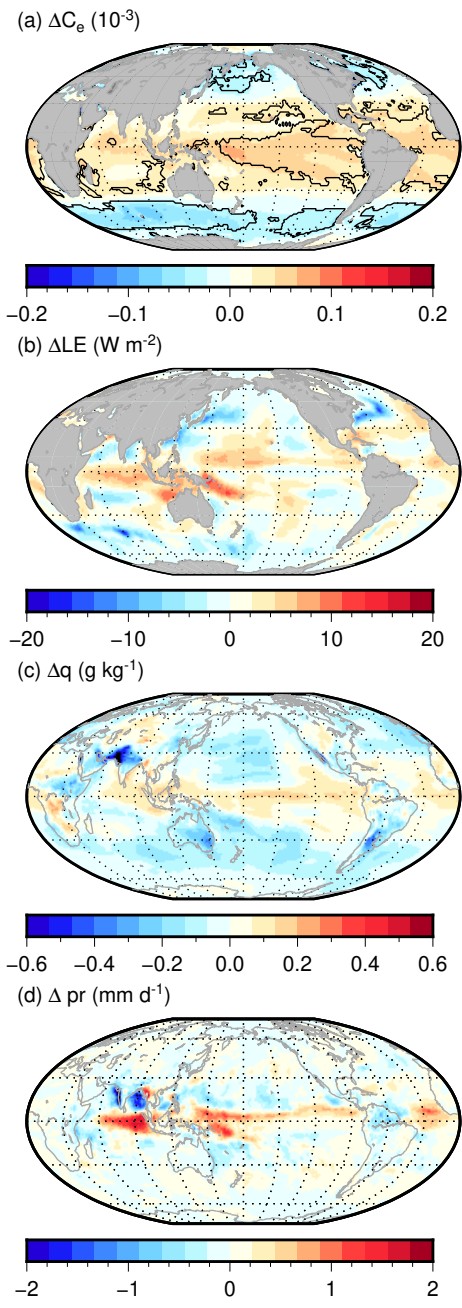

**Figure 17.** Maps of the annual mean differences for $C_e$ (a), $LE$ (b), $q$ at the lowest atmospheric level (c) and daily precipitation $pr$ (d) between WASP and ECUME in AMIP simulations over the period 1979–2014. The black lines indicates the zones where the difference is significant with respect to the interannual variability (Student test at 95 % uncertainty).



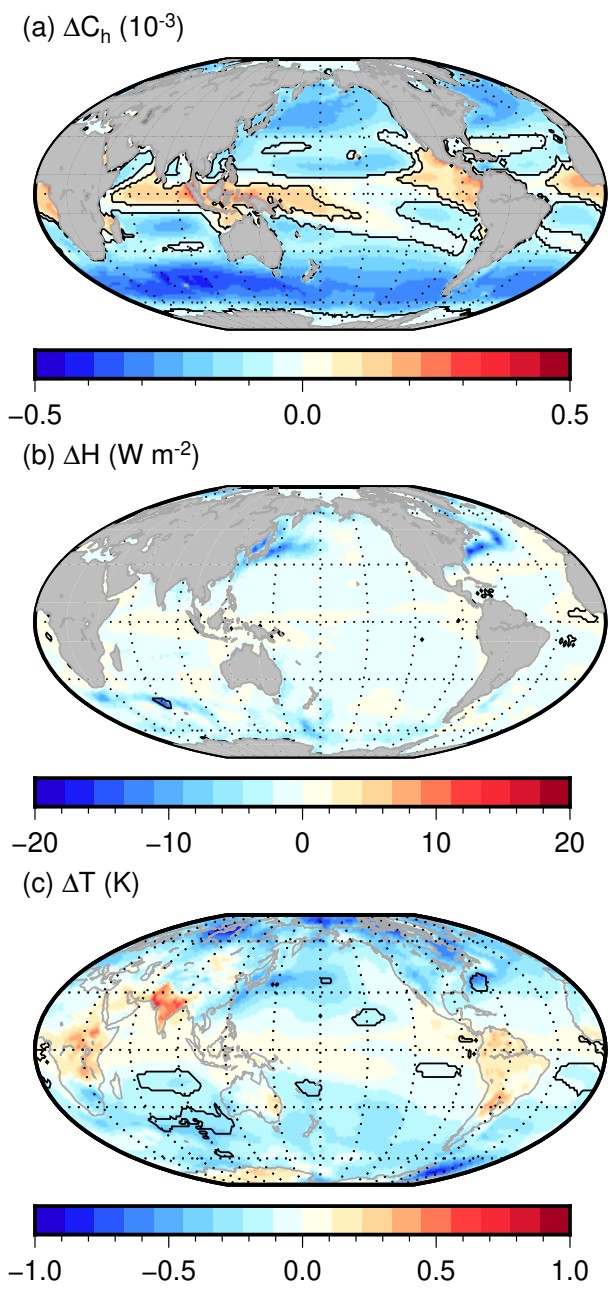

**Figure 18.** Maps of the annual mean differences for $C_h$ (a), $H$ (b), and $T$ at the lowest atmospheric level (c) between WASP and ECUME in AMIP simulations over the period 1979–2014.

turbulence on the cold side of the front is decoupled between the upper MABL and the surface and the surface wind is reduced.

In the HPE that occurred in West Mediterranean in October 2016, the change of parameterization affects the strong, moist

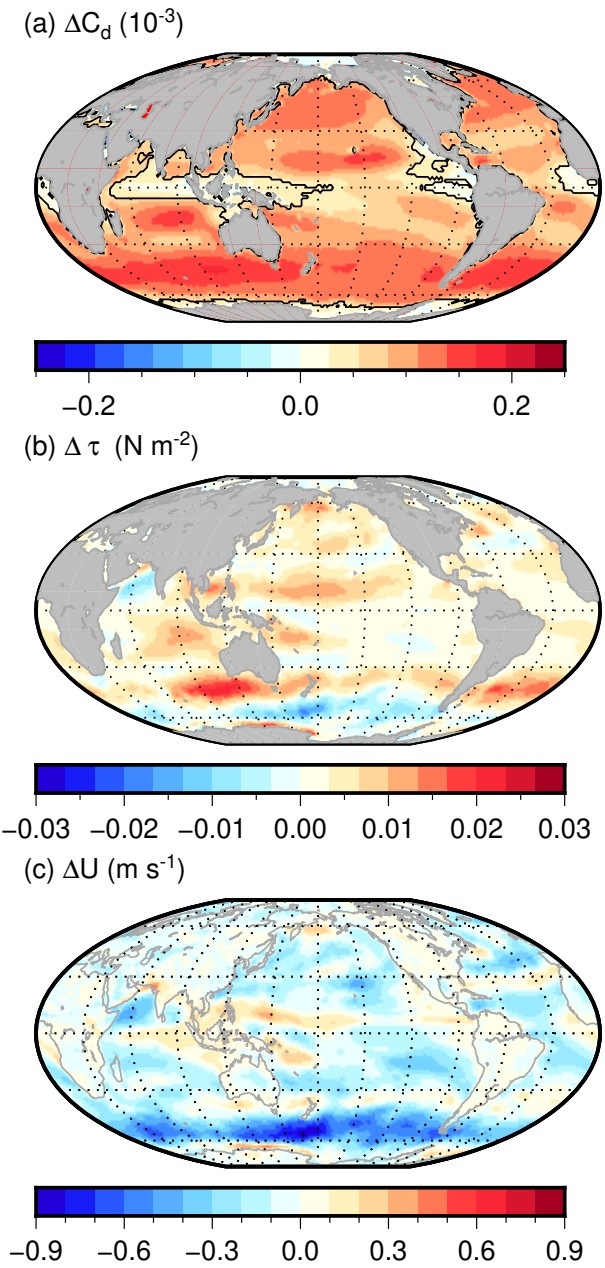

**Figure 19.** Maps of the annual mean differences for $C_d$ (a), $\tau$ (b), and $U$ at the lowest atmospheric level (c) between WASP and ECUME in AMIP simulations over the period 1979–2014.

low-level flow leading to change in heavy precipitation through dynamical effects mainly. Representing the surface fluxes by WASP rather than ECUME increases the surface roughness and decreases the turbulent heat fluxes. It results in a slightly less





intense but more stationary convective system at sea without significant impact on the precipitation forecast. Validating the parameterization in cyclonic conditions is an important step towards its use for operational forecast. In the present case, it also helped to adjust the heat transfer coefficients above 20 m s$^{-1}$, where observations provide no constrain anymore. Several case studies in the South West Indian Ocean basin showed that the intensity of cyclones is slightly reduced with respect to ECUME, mainly due to the decrease of the enthalpy transfer coefficient in case of strong and cyclonic winds. Finally, testing the impact of the change on a climatic atmosphere-only simulation gives results consistent with existing studies. The combined effects of the changes of the wind stress and heat fluxes enhance moisture extraction and precipitation in the intertropical zone whereas lower atmosphere is drier and cooler at midlatitudes. This work is a first step towards further development of parameterization of both momentum and turbulent heat fluxes. Ongoing work aims at refining the representation of the variability of the fluxes possibly due to sea-state variations, including the effects of wave breaking, and the effect of sea spray on the momentum and heat fluxes. Indeed, the effect of sea spray, though likely significant for both the momentum and heat transfer in breaking conditions, is not considered in WASP. Recent and ongoing studies aim at building droplets source functions more consistent with the (few) existing observations for large droplets, meant to affect the turbulent fluxes (Bruch et al., 2021), and the corresponding parameterization of their impact on the fluxes following Bao et al. (2011).

## Appendix A: WASP definition

### A1 Transfer coefficients

In WASP, the Charnock parameter $\alpha$ is defined differently depending on the wind speed range, as follows:

- 10 m wind speed $U_{10}$ below 7 m s$^{-1}$ is a power of $U_{10}$: $\alpha = aU_{10}^b$, where a = 0.7 and b = $-2.52$;

- when $U_{10}$ is above 7 m s$^{-1}$, the dependency on wave age $\chi = c_p/U_{10}$ is introduced and is defined as $\alpha = A\chi^B$, where A and B are polynomial functions of $U_{10}$.

$$\begin{cases} A & = & A_0 + A_1 U_{10} + A_2 U_{10}^2 + A_3 U_{10}^3 \\ B & = & B_0 + B_1 U_{10} + B_2 U_{10}^2 + B_3 U_{10}^3, \end{cases} \tag{A1}$$

as detailed in Table A1.

Thus, the dependency of the Charnock parameter and the decrease in the drag coefficient under very strong wind conditions are represented, and the WASP parameterization, unlike those based on wave-age Charnock parameters, is suitable for very high wind speeds.





**Table A1.** Coefficients of the polynomial functions defining the WASP Charnock parameter depending on the wind speed.

| | $A_0$ | $A_1$ | $A_2$ | $A_3$ |
| --- | --- | --- | --- | --- |
| | $B_0$ | $B_1$ | $B_2$ | $B_3$ |
| $7 \leq U_{10} < 23$ | $-9.202$ | $2.265$ | $-1.34 \times 10^{-1}$ | $2.35 \times 10^{-3}$ |
| | $-4.12 \times 10^{-1}$ | $-2.225 \times 10^{-1}$ | $1.178 \times 10^{-2}$ | $1.616 \times 10^{-4}$ |
| $23 \leq U_{10} < 25$ | $2.27$ | $6.67 \times 10^{-2}$ | $0$ | $0$ |
| | $-2.41$ | $4.30 \times 10^{-2}$ | $0$ | $0$ |
| $U_{10} > 25$ | $9.81 \times 10^{-2}$ | $-4.13 \times 10^{-3}$ | $4.34 \times 10^{-5}$ | $1.16 \times 10^{-8}$ |
| | $0$ | $0$ | $0$ | $0$ |

## A2 Stability functions

The stability functions for momentum and heat fluxes are taken as in Beljaars and Holtslag (1991), modified to be implemented in the COARE 3.0 algorithm (Fairall et al., 2003). In unstable conditions, the stability function for momentum is:

$$\Psi_M = 2\log\left(\frac{1+x}{2}\right) + \log\left(\frac{1+x^2}{2}\right) - 2\tan^{-1}(x) + \frac{\pi}{2} \tag{A2}$$

with $x = (1 - 15z/L)^{1/4}$, and in conditions of free convection:

$$\Psi_M = 1.5\log\left(\frac{y^2+y+1}{3}\right) + \sqrt{3}\tan^{-1}\left(\frac{2y+1}{\sqrt{3}}\right) + \frac{2\pi}{\sqrt{3}} \tag{A3}$$

with $y = (1 - 10.15z/L)^1/3$ and in stable conditions:

$$\Psi_M = -1\left[1 + z/L + \frac{2}{3}\frac{(z/L - 14.28)}{e^c} + 8.5\right] \tag{A4}$$

with $c = 0.35z/L$.

The stability function for heat or humidity is defined as:

$$\Psi_H = 2\log\left(\frac{1+x^2}{2}\right) \tag{A5}$$

with $x = (1 - 15z/L)^{1/2}$, in conditions of free convection:

$$\Psi_H = 1.5\log\left(\frac{y^2+y+1}{\sqrt{(3)}}\right) + \frac{2\pi}{\sqrt{(3)}} \tag{A6}$$

with $y = (1 - 34.15z/L)^1/3$ and in stable conditions:

$$\Psi_H = -1\left[\left(1 + \frac{2}{3}\frac{z}{L}\right)^{1.5} + \frac{2}{3}\frac{(z/L - 14.28)}{e^c} + 8.525\right] \tag{A7}$$

with $c = 0.35z/L$.





**Appendix B: Detail of datasets used for validation**

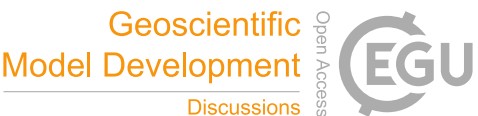

**Table B1.** Eddy covariance datasets used for fitting the neutral transfer coefficients in wind speed range 0–20 m s$^{-1}$. All these measurements except CBLAST low (RP *FLIP*) were made onboard ships and most of them were used for the COARE 3.5 parameterization for wind stress Edson et al. (2013)

| Campaign | Year | Height | Wind range | Sampling | Reference |
| --- | --- | --- | --- | --- | --- |
| | | m | m s$^{-1}$ | min | |
| **ASTEX** | 1992 | 21 | 3–10 | 30 | Albrecht et al. (1995) |
| **CAPRICORN** | 2016 | 21 | 1–16 | 10 | Bharti et al. (2019) |
| **CBLAST low** | 2001–2003 | 12 | 0–17 | 20 | Edson et al. (2007) |
| **DYNAMO** | 2011–2012 | 15.6–17.75 | 0–14 | 10 | Moum et al. (2014), De Szoeke et al. (2015) |
| **FASTEX** | 1996–1997 | 15.5–17.7 | 1–18 | 10 | Hare et al. (1999), Joly et al. (1999) |
| **HIWINGS** | 2013 | 14–15.9 | 1–19 | 10 | Blomquist et al. (2017) |
| **JASMINE** | 1999 | 14.8–17.7 | 0–13 | 10 | Fairall et al. (2000) |
| **KWAJEX** | 1999 | 15.5–17.7 | 0–9 | 10 | Fairall et al. (2003) |
| **MOORINGS** | 1999 | 15.5–17.7 | 0–13 | 10 | Fairall et al. (2003) |
| **NAURU** | 1999 | 15.5–17.7 | 0–10 | 10 | Fairall et al. (2003) |





**Table B2.** Additional EC datasets used for fitting the neutral transfer coefficients in wind speed range 0–30 m s$^{-1}$. ** indicates that measurements of enthalpy fluxes are available.

| Campaign | Year | Height m | Wind range m s$^{-1}$ | Platform | Reference |
|---|---|---|---|---|---|
| **Halifax** | 1976 | 12 | 8–22 | platform | Smith (1980) |
| **Halifax** | 1976 | 12 | 4–24 | platform | Large and Pond (1981) |
| **HEXOS**** | 1986 | 10–18 | 6–23 | platform | DeCosmo et al. (1996) |
| **HEXOS** | 1986 | 6 | 7–20 | platform | Janssen (1997) |
| **BaltEx**** | 1998 | 10,18 | 6–18 | platform | Rutgersson et al. (2001) |
| **RASEX** | 1994 | 3 | 4–15 | platform | Fairall et al. (2003) |
| **South China Sea**** | 2010 | 20 | 0–22 | platform | Zou et al. (2017) |
| **SWADE** | 1990 | 12 | 4–14 | ship | Donelan et al. (1997) |
| **ITOP** | 2010 | 5.4 | 3–28 | buoy | Potter et al. (2015) |
| **CBLAST high**** | 2003 | 70–370 | 17–29 | air | Black et al. (2007), French et al. (2007), Zhang et al. (2008) |
| **GOTEX** | 2004 | 30–50 | 11–20 | air | Romero and Melville (2010) |
| **GFDex**** | 2007 | 36–43 | 15–19 | air | Petersen and Renfrew (2009) |
| **British Iles**** | 2007–2013 | 35–80 | 4–25 | air | Cook and Renfrew (2015) |



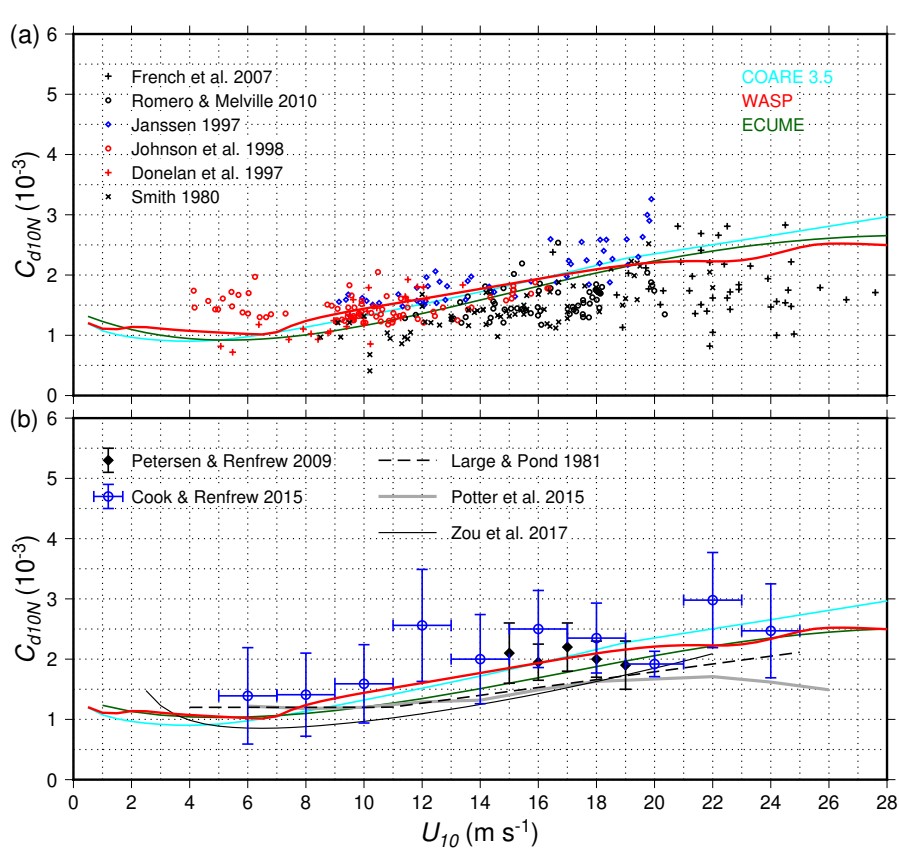

**Figure B1.** Neutral drag coefficient with respect to 10 m wind speed for different parameterizations (COARE3.5, Edson et al. (2013); COARE3.0, Fairall et al. (2003); ECUME Roehrig et al. (2020) and WASP) in comparison with additional observations (see Table B2) up to $28 \, \mathrm{m \, s^{-1}}$.





**Table B3.** Additional, indirect datasets used for fitting the neutral transfer coefficients in wind speed range above 30 m s$^{-1}$. ** indicates that measurements of enthalpy fluxes are available

| Reference | Method | Wind range m s$^{-1}$ |
|---|---|---|
| Powell and Ginis (2006) | Dropsondes | 27–62 |
| Richter and Stern (2014) ** | Dropsondes | 20–50 |
| Vickery et al. (2009) | Dropsonde & modelling | 18–54 |
| Bell et al. (2012) ** | SAMURAI | 54–72 |
| Jarosz et al. (2007) | Inversion of surface currents | 20–47 |
| Sanford et al. (2011) | Inversion of oceanic resp. | 22–47 |
| Hsu et al. (2017) | Inversion of oceanic resp. | 27–57 |

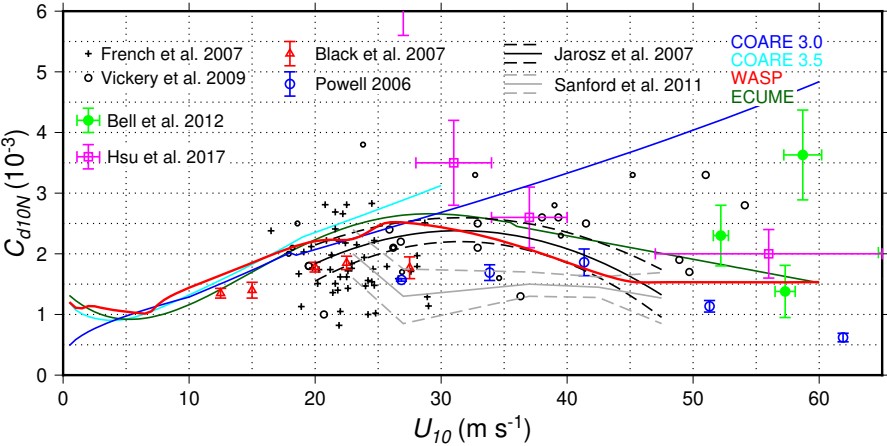

**Figure B2.** Neutral drag coefficient with respect to 10 m wind speed for different parameterizations (COARE3.5, Edson et al. (2013); COARE3.0, Fairall et al. (2003); ECUME Roehrig et al. (2020) and WASP) and detail of observations up to 60 m s$^{-1}$ (see Tables B2 and B3).





## Appendix C: Examples of SURFEX v8.1 namelists using WASP

### C1 WASP without wave impact





**Table C1.** SURFEX namelist (EXSEG1.nam) parameters used for using WASP without wave impact.

| $NAM_SEAFLUXN | |
|---|---|
| CSEA_FLUX | 'WASPV3' |
| LWAVEWIND | .TRUE. |



## C2 WASP with wave impact





**Table C2.** SURFEX namelist (EXSEG1.nam) parameters used for using WASP with wave coupling, with a coupling frequency of 600 s.

| $NAM_SEAFLUXN | |
|---|---|
| CSEA_FLUX | 'WASPV3' |
| LWAVEWIND | .FALSE. |
| $NAM_OASIS | |
| LOASIS | .TRUE. |
| LOASIS_GRID | .TRUE. |
| CMODEL_NAME | 'mesonh' |
| $NAM_SFX_WAVE_CPL | |
| CWAVE_U10 | 'MNH__U10' |
| CWAVE_V10 | 'MNH__V10' |
| CWAVE_CHA | '          ' |
| XTSTEP_CPL_WAVE | 600.0 |
| CWAVE_UCU | '          ' |
| CWAVE_VCU | '          ' |
| CWAVE_HS | 'MNH___HS' |
| CWAVE_TP | 'MNH___TP' |



*Code and data availability.* The new developed code will be included in the next official version of SURFEX v9.0. The complete code of SURFEX v8.1 including WASP and the WASP subroutines is available at https://doi.org/10.5281/zenodo.4557378 (Bouin, 2021); the data used to tune the transfer coefficients are available by contacting C.W. Fairall (NOAA Physical Science Division) or J.B. Edson (U. Conn. Marine Sciences). The offical release of SURFEX v8.1 offline is available at https://www.umr-cnrm.fr/surfex/spip.php?rubrique141.

The ARPEGE-Climat model is only available to registered users for research purposes. The access to the AROME code is ruled by the first Memorandum of Understanding of the ACCORD consortium (http://www.accord-nwp.org). For non-commercial research purposes, AROME can be distributed upon signature of a licence agreement (see http://www.accord-nwp.org/?ACCORD-MoU-2021-2025 for conditions). The modifications of the ARPEGE and AROME codes including WASP and used for the case studies of this paper are stored at https://doi.org/10.7910/DVN/WXZ1FO. The output parameters of the simulations used to validate WASP (Figures 8 to 19) and the data of

the Lion buoy used to compare the wave effects in Figure 5 are available at https://doi.org/10.5281/zenodo.6783319. The SAR product was obtained from Ifremer/Cyclobs and produced with SAR wind processor co-developped by IFREMER and CLS and can be accessed here: https://cyclobs.ifremer.fr/app/archive/2022/SI/sh082022 (last access on 30/06/2022). The Best Track data have been extracted from the Best Track data base of the Direction Régionale de l'Océan Indien (DIROI) of Météo-France. These data are shared with the IBTracs data base (https://www.ncdc.noaa.gov/ibtracs/) after a subjective reanalysis by the DIROI forecasters at the end of each TC season.

*Author contributions.* MNB designed and implemented the WASP parameterization, and performed the Iroise case study. CLB and CS designed and ran the Mediterranean case study. SM designed and performed the tropical cyclones case studies. AV designed and performed the climate-scale simulation. All authors contributed to the writing and revising of the manuscript.

*Competing interests.* The authors declare that they have no conflict of interests

*Acknowledgements.* The authors acknowledge the Pôle de Calcul et de Données Marines for the DATARMOR facilities (storage, data access,
computational resources). We are also grateful to C. Fairall for providing field measurements. We thank A. Mouche and O. Archer (LOPS) and CLS for providing the SAR products in the framework of the Ifremer/Cyclobs project.





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
