# Peer review of "The wave-age dependent stress parameterization (WASP) for momentum and heat turbulent fluxes at sea in SURFEX v8.1."

_Geoscientific Model Development, 2023_

## Author Comment (AC1)

**The wave-age dependent stress parameterization (WASP) for momentum and heat turbulent fluxes at sea in SURFEX v8.1 by Bouin et al.**
**Response to reviewers**

The authors thank both reviewers for their time and comments that will help to significantly improve this paper. We recognize that some parts lacked precision and we hope that the new version will be more accurate and clearer about the goal of this work and its limitations.

Reviewer 1 (C. Fairall)
This paper describes a package of air-sea flux parameterizations called WASP as executed in SURFEX v8.1. The idea is to update to parameterizations of momentum, sensible, and latent heat based on a collection of direct and indirect observations taken from the literature. Because the transfer coefficients are considered well-constrained by tens of thousands of direct observations for wind speeds 0-25 m/s, the authors emphasize the wind speed regime greater than 25 m/s. So, WASP is a good fit to existing parameterizations such as COARE in the 0-25 m/s range. The authors attempt to reach a crude consensus representation of the data for U>25 m/s. The approach is to throw everything they can find (uncritically) into wind speed bins and produce a mean fit, sort of. The new package is then run in several different models for mesoscale weather and global climate model implementations. The results from WASP and other competing models are compared.

The authors face a well-known problem – the existing high-quality data describe the fluxes for U<25 m/s but modeling centers and researchers insist on running their models in hurricanes and other high wind speed events well outside the constraint boundaries. We have to provide them with our best estimates of the transfer coefficients regardless of the quality of the observations. Yes, for U>25 m/s we have a collection of bad or uncertain data, but the average of bad = good, right? It is easy to quibble with details of what the authors have done. For example, Black et al 2007 and French et al 2007 are the same data. Or, why include data from 1976? Or, if a data set's value for Cd is 25% below COARE at U=23 m/s, why would you trust if for U=28 m/s? Or, if WASP if a good representation of Cd for U>45 m/s, why to do 8 data points fall above WASP and only 3 points fall below in Fig. 2B? Reasonable quibbles, but a glance at Figs. 2 and 2b suggests it makes no sense for this reviewer to second guess the authors. Is WASP better than just capping Cd at U=30 m/s? I have no idea.

We feel that there is maybe a misunderstanding about the aim of this work. WASP is intented to feel the gap between the two bulk parameterizations commonly used in SURFEXV8.1: COARE3.0, enabling coupling with waves but with some lack of variability due to waves and a necessary capping above 30 m/s, and ECUME, which cannot be coupled with waves. We do not pretend to obtain a better parameterization with WASP than with an adaptation of COARE3.0 or COARE3.5. WASP simply meets the needs of variability due to waves and mean behavior close to the available observations up to 60 m/s, and is usable in the Météo France operational systems. This was already mentioned in the paper (l. 108-110). We made this even clearer in the new version of the paper, at the beginning of Section 4.

So, on balance I recommend publication of this paper but with some substantial changes to address problems with the manuscript. My more important issues are

1. The presentation of the data and the various figures is a bit chaotic. I have trouble figuring out exactly which data went into which figures. Perhaps there are too many such figures?

The authors state using a data base of 27,000 values for C3, 21,000 for Ch, and 24,000 for Ce and refer to Table B1. But, the data in Table B1 are likely less than 1,000 points. And none of the data in Table B2 (way less than 20,000 observations) were used in COARE fits. So where do the 72,000 values come from?

Thank you for this comment. Indeed, the presentation of the datasets was not clear due to the use of (probably) too many figures. We changed this by gathering Fig. 3a and 4a in one Figure (3) and replacing Fig. B1 by the combination of Fig. 3b and 4b. About the number of observations used, we feel that there is maybe a misunderstanding about what we called "observations": in the present case, 10 to 30 min bins of high frequency measurements used for flux computations by EC (see Table B1). For instance, the CAPRICORN dataset corresponds to 3231 observations, CBLAST to more than 3800, etc. The total is actually above 27,000 for the Cd values.

2. The authors need to make it clear that the use of the Gulf of Lion buoy data and the case study intercomparisons model outputs in section 3 are not actual **validation** but are model intercomparisons. There are no actual direct flux observations involved.

Yes, again, we think that maybe the objectives of these tests was not clear: what we want to check here is that the use of WASP in realistic and already validated case studies (by comparison with observations for case studies of sections 3.1, 3.2 and 3.3) give results close to or slightly better than the previously used parameterizations. Observations (SAR winds) are actually used in the TC case study, so we think what we can keep the validation term in this case. For the other test runs, we use "comparison with previous simulations" instead, and we hope it describes more accurately the work done.

3. I have my doubts about the value of the comparisons done in Section 3. Do they provide a basis for deciding WASP is superior to ECUME or ERA5?

Again, we do not claim that WASP is superior to the previous parameterizations used in SURFEXV8.1. Its development was necessary for operational and research applications because neither ECUME nor COARE3.0 permitted to couple wind and waves with a good level of variability and a close fit to observations in cyclonic conditions. We made this clearer in the new version.

4. The authors make some claims about the change in processes that lead to the saturation and decrease of Cd with increasing wind speed. They state in several places that this is due to the transition to a state with no 'wave growth' because of a balance with wave breaking. The is a difference between wave energy and momentum input from the atmospheric (via the pressure-slope correlation) versus increase in the actual wave energy (increasing significant wave height) which you might characterize as 'growth'. It is doubtful there is a balance between pressure input and dissipation in hurricanes (which have relatively young waves) but even if there is it does not require Cd to level off or decrease. If the authors wish to invoke a mechanism, I suggest they spend some time to make a clear and defensible argument.

In fact, the process we meant by dissipation was wave breaking, which is generally present for surface wind speed above 7 m/s. Since this point was also raised by Reviewer 2, we added a few sentences: According to previous work, the physical mechanisms likely to explain the observed saturation or decrease of the drag coefficient above 30 m s−1 are air-flow separation due to wave breaking Kudryavtsev et al. (2014), changes in the wind profile close to the surface due to high concentration of sea-spray Andreas (2004), or inclusion of non-linear effects in the critical layer theory for wave-growth Miles (1957) with explicit

calculation of the momentum transferred to capillary-gravity waves Janssen and Bidlot (2023).

5. The use of the oceanic momentum budget data sets given in Table B3 might be worth pondering. There is a difference in the momentum removed from the atmosphere (say, inferred from dropsonde on measured with sonic anemometers) vs the momentum realized in ocean currents and/or input to waves.

Yes, thank you for this comment. These data sets were included because we wanted to take advantage of all the information available for wind speeds above 40 m/s. We agree that the uncertainties on these estimates is likely much larger those on direct, atmospheric measurements, but this should be at least partly reflected by the larger errors bars (Fig. B2) We added a sentence to discuss this point.

Here are a few other editorial comments

Eqs. 1 and 2. The authors have ignored gustiness (distinction between mean wind speed and magnitude of the mean wind vector) in their parameterization. This leads to Cd, Ch, and Ce becoming singular as wind speed approaches zero.

Yes, thank you for this comment. The gustiness is present in WASP as in all the SURFEXV8.1 bulk parameterization but not included in the Eq 1 and 2. We make this clear in the text.

Fig. 1 Doesn't it bother the authors that WASP and COARE curves fall well below the average of the data for U<8 m/s? Is this a consequence of ignoring gustiness? Why are there more black data than blue data?

Yes, this is a very good comment. Indeed, this is due to ignoring the effect of gustiness in the offline parameterization.

The black symbols correspond to mean/standard deviation and are represented as soon as 5 or more individual data are present, while the blue boxes and whiskers correspond to the 10% and 90% quantiles, and are shown when more than 10 data are present in the bin. This explains why boxes and whiskers are not shown for the highest winds. This is now clear in the caption.

Figs. 2 and 3 I don't see the COARE 3.0 line.

COARE3.0 and WASP CHN and CEN are almost superimposed in Fig. 3 and 4. We now specify it in the caption.

Fig. 5. That caption is confusing. I think the dots in 5a are the wave dependent COARE3.5. The green line is misidentified. A good figure because it shows the Oost and Taylor wave formulae have problems.

Yes, thank you for that, the caption is indeed wrong is several respects. This has been corrected: COARE3.5 is now shown in black, and the variability with wave age in (a) corresponds to WASP.

Line 199 Not sure WASP dependence on wave age is shown.

Correct. It was actually shown in Fig 5a (but with a misleading caption) but not in Fig. 1. This has been corrected (Fig. 4a).

Fig. 7. This figure is confusing and hard to interpret. Nice try, but it doesn't work. Suggest you find something simpler.

Yes, as you said, it was a tentative to represent too much information in a single plot. We changed it for a simpler version and we hope it is now easier to understand.

Line 256  What is 'wave effect dependent on wind speed only'?

WASP is designed to be used preferably in coupled mode with a wave model, but it can be used in stand alone mode. In this case, the wave age is computed as a function of the 10 m wind speed. We reformulated.

Line 310-318  This text is repeated in 318-327.

Corrected, thank you.

Table B1.  I doubt these data were used by Edson et al. 2013.

Correct, thank you. This has been corrected.

Perhaps compare WASP to the GFDL parameterization of Cd in

Chen, Xuanyu, Tetsu Hara, and Isaac Ginis. 2020. "Impact of Shoaling Ocean Surface Waves on Wind Stress and Drag Coefficient in Coastal Waters: 1. Uniform Wind." *Journal of Geophysical Research: Oceans* 125 (7): e2020JC016222. doi:10.1029/2020JC016222.

Thank you for the suggestion and reference. We added the GFDL parameterization in the comparison in Fig.2 and a comment on that in the text. Note that, for very strong winds, GFDL is close to ECUME.

Reviewer 2

Very interesting paper, giving a comprehensive summary of the work done to developed WASP parameterisation.

Thank you.

Here are my main comments:

Line 67: "the dependence of the drag coefficient on the wave growth is not relevant for wind speeds higher than 30 m/s". Whereas I agree that for very strong wind conditions, other physical processes might become relevant than for calmer condition, I would not state that "the dependence of the drag coefficient on the wave growth is not relevant for wind speeds higher than 30 m/s". Recent work by Janssen and Bidlot (2023) showed that aspect of the wind-wave interaction for strong winds become a bit more subtle, with direct nonlinear feedback on the waves on the growth by winds, therefore a direct impact on the momentum exchange at the surface.

Peter A.E.M. Janssen and Jean-Raymond Bidlot, 2023. Wind–Wave Interaction for Strong Winds" (10.1175/JPO-D-21-0293.1). Journal of Physical Oceanography, vol. 53, no. 3. https://journals.ametsoc.org/view/journals/phoc/53/3/JPO-D-21-0293.1.xml

Thank you for the comment and the reference. We changed the text accordingly: According to previous work, the physical mechanisms likely to explain the observed saturation or decrease of the drag coefficient above 30 m s−1 are air-flow separation due to wave breaking Kudryavtsev et al. (2014), changes in the wind profile close to the surface due to high concentration of sea-spray Andreas (2004), or inclusion of non-linear effects in the critical layer theory for wave-growth

Miles (1957) with explicit calculation of the momentum transferred to capillary-gravity waves Janssen and Bidlot (2023).

Line 69: I agree for the need to reproduce the saturation/decrease of the drag coefficient for strong winds. As a motivation for the extension of the wave generation theory of Janssen (Janssen and Bidlot 2023) (to be implemented in 2024), a simple empirical adjustment of his original work was implemented a few years ago (2020) in the coupled system at ECMWF. The impact of which was essential in obtaining decent forecast results in global simulations of tropical cyclones at high resolution (for a global system, 4.4 km) (Majumdar et al. 2023)

Majumdar, S. J., L. Magnusson, P. Bechtold, J. Bidlot, and J. D. Doyle, 2023:Advanced tropical cyclone prediction using the experimental global ECMWF and operational regional COAMPS-TC systems.Mon. Wea. Rev., https://doi.org/10.1175/MWR-D-22-0236.1, in press.

Thank you. We added the reference, and the sentence now reads: The saturation or decrease observed for cyclonic wind speeds must be reproduced in a parameterization (using an analytical function or capping) to match the observations and enable more realistic simulation of the tropical cyclone intensity Majumdar et al. (2023).

Line 70: the statement that the heat exchange coefficient shows no clear dependence on the wind speed, nor the sea state is a bit misleading as later with equation (7), a dependence on z0 is used, which has been shown to have such dependence. I know this results in a weak dependence and (7) comes about from scaling arguments and the experimental results have large scatter.

Indeed, we changed the text accordingly:

Line 105: Note that Janssen (2004) makes the point that the wave age parameter should be defined with the friction velocity (u*) rather than U10 since u* will vary even when U10 is constant. Since (6) is a bulk parameterisation fitted to a cloud of experimental data, it is probably a fair assumption.

Yes. Actually, we balanced between using u* or U10 and we decided for U10 because the computation of the wave age can be done outside of the iterative loop in the code, and because we checked that it makes no significant difference in the Cd values. We now mention it in the text.

Line 130: Cp in (6) is for the peak of the windsea spectrum. Do you make sure to only consider the windsea spectrum when determining the wave peak period (Tp) from which Cp will be determined, otherwise Tp might correspond to the swell part of the spectrum that has nothing to do with the local wind generation?

Yes, this is a very good point. Indeed, we use the peak period of the windsea, the parameters sent from WW3 to OASIS have been modified accordingly. We now make this clear in the text.

Line 190: Please refer to Janssen and Bidlot (2023) for a novel modelling of the high frequency gravity-capillary spectrum.

Done.

Line 194: why would the coupling via the Charnock coefficient prevent the comparison of wave parameters with observations?

Indeed, it would not. What we meant here was that the parameters produced by wave models, and easily comparable against observations, are significant wave height and peak period. Wave models (especially those used in opertaional systems) are designed and tuned to produce Hs and Tp as close

as possible to observations. In other terms, a lot of efforts is put by model developpers into these parameters, and we feel that Hs or Tp are the best parameters to ensure that the variability of sea state is well represented and to validate (as an intermediate step) this predicted sea state against observations. We reformulated this part of the sentence.

Line 195, (ii): I disagree, this is just parameter fitting to observed data sets.

Of course, the mean values of the drag coefficient are fitted to observations. What we meant here was that the variability of the drag with wave age, for a given wind speed, corresponds to the physical process of momentum absorbed by wave growth, and that this variability is limited to a given range of wind speed. We reformulated.

Line 309: which version of WW3?

WW3 v5.16, as now specified in the text.

Minor corrections:

Line 49: ratio of the wave peak period to the near-surface wind -> ratio of the wave phase speed for the peak of the wave spectrum to the near-surface wind

Corrected, thank you.

Line 292: in link -> in line

Corrected, thank you.

Lines 315-327 are repeated.

Corrected, thank you.

Line 399 and 401: IFS -> operational ECMWF forecasts

Changed.

Line 445: whatever the wind speed below -> wherever the wind speed is below

Changed.